# Cost-Aware Routing for Efficient Text-To-Image Generation

**Qinchan (Wing) Li**$^\diamond$     *ql840@nyu.edu*

**Kenneth Chen**$^\diamond$     *kc4906@nyu.edu*

**Changyue (Tina) Su**$^\diamond$     *cs7483@nyu.edu*

**Wittawat Jitkrittum**$^{\dagger*}$     *wittawatj@gmail.com*

**Qi Sun**$^\diamond$     *qisun@nyu.edu*

**Patsorn Sangkloy**$^{\diamond*}$     *patsorn.sangkloy@gmail.com*

$\diamond$ *Tandon School of Engineering, New York University*
$\dagger$ *Google, New York*

**Reviewed on OpenReview:** *https://openreview.net/forum?id=Jbe9AVsYS6*

## Abstract

Diffusion models are well known for their ability to generate a high-fidelity image for an input prompt through an iterative denoising process. Unfortunately, the high fidelity also comes at a high computational cost due to the inherently sequential generative process. In this work, we seek to optimally balance quality and computational cost, and propose a framework to allow the amount of computation to vary for each prompt, depending on its complexity. Each prompt is automatically routed to the most appropriate text-to-image generation function, which may correspond to a distinct number of denoising steps of a diffusion model, or a disparate, independent text-to-image model. Unlike uniform cost reduction techniques (e.g., distillation, model quantization), our approach achieves the optimal trade-off by learning to reserve expensive choices (e.g., 100+ denoising steps) only for a few complex prompts, and employ more economical choices (e.g., small distilled model) for less sophisticated prompts. We empirically demonstrate on COCO and DiffusionDB that by learning to route to nine already-trained text-to-image models, our approach is able to deliver an average quality that is higher than that achievable by any of these models alone. Code is available at https://github.com/winglicopy/CATImage.

## 1 Introduction

While diffusion models have set a new standard for photorealism in generative art (Ho et al., 2020), their operational costs remain a major challenge. The generation of a single image can involve many denoising steps, each utilizes a learned denoiser model with potentially over a billion parameters (Rombach et al., 2022). This makes in-the-wild adoption (i.e., on-device) challenging and raises valid concerns about their environmental sustainability (Sarah Wells, 2023; Kate Crawford, 2024; Kaack et al., 2022). To address this, a significant body of research has explored optimization strategies such as network simplification (Li et al., 2024; 2023a) and model distillation (Sauer et al., 2024; Salimans & Ho, 2022; Meng et al., 2023; Liu et al., 2023).

However, these existing methods typically apply the same degree of optimization irrespective of the task's intrinsic difficulty. This results in a single model with a fixed computational cost, which is inherently suboptimal as the generative effort required to synthesize an image varies with the complexity of the input prompt. For example, a simple prompt like *a white and empty wall* requires fewer denoising steps to generate a high-quality image than a complex one like *a colorful park with a crowd*, as shown in Figure 1.

---

*Now at Eigen 4D Inc. (https://eigen4d.com).

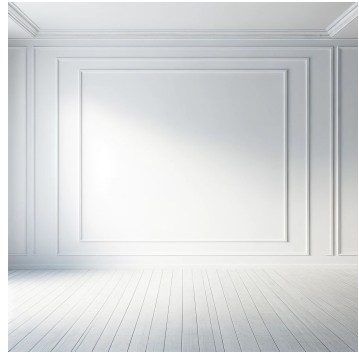
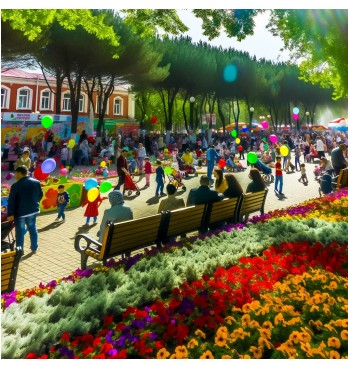
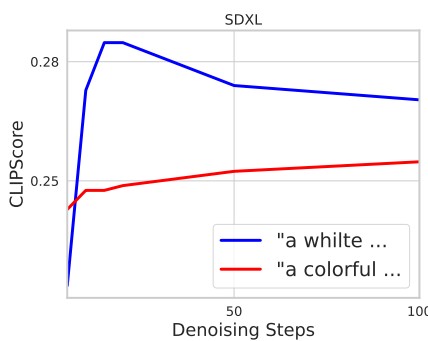

(a) *'a white and empty wall'*     (b) *'a colorful park with a crowd'*     (c) *quality trends across numbers of steps*

Figure 1: *Two input prompts that require different denoising steps to ensure quality.* As shown in Figure 1c, prompt Figure 1a only requires a small number of denoising steps to reach a high CLIPScore. By contrast, the more complex prompt Figure 1b requires over 100 steps to reach a similar quality. Key to our proposed *CATImage* is to allocate an appropriate amount of computation for each prompt, so that the overall computational cost is reduced while the quality remains the same.

With the motivation to adaptively allocate computational budget, we present *CATImage*, a framework that allows the amount of computation for text-to-image generation to vary for each prompt. Our framework operates with a pre-defined set of choices that can be chosen adaptively for each input prompt. Each choice represents a text-to-image generation function and has a distinct profile of computational cost and the expected image quality. Concretely, these choices may correspond to different numbers of denoising steps of the same diffusion model (i.e., homogeneous choices), disparate, independent text-to-image generative models (i.e., heterogeneous choices), or a combination of both. The proposed *CATImage* aims to adaptively select the right choice (i.e., "routing") for each input prompt, in such a way that expensive choices (e.g., 100+ denoising steps) are only for complex prompts. Our approach enables a joint deployment of diverse text-to-image models and has a potential to deliver higher average image quality compared to using any individual model in the pool, while allowing the average computational cost to be adapted at deployment time.

In summary, our contributions are as follows.

1. We precisely formulate a constrained optimization problem for the above routing problem (Section 3.1). The formulation aims to maximize average image quality subject to a budget constraint on the generation cost.

2. We study the theoretically optimal routing rule that optimally trades off the average quality and cost (Section 3.2). Based on the optimal rule, we construct a plug-in estimator that can be trained from data.

3. We perform a series of objective analyses on the COCO (Lin et al., 2014) and DiffusionDB datasets (Wang et al., 2022b). Our findings show that, through adaptive routing, our proposal matches the quality of the largest model in the serving pool (namely, Stable Diffusion XL from Radford et al. (2021) with 100 denoising steps) with only a fraction of its computational cost (Table 3).[1]

## 2 Background: Text-To-Image Generative Models

Let $\mathbf{x} \in \mathcal{X}$ denote an input text prompt, and $\mathbf{i} \in \mathcal{I} \doteq [0,1]^{W \times H \times 3}$ denote an image described by the prompt, where $W, H \in \mathbb{N}$ denote the width and the height of the image (in pixels), and the last dimension denotes the number of color channels. A text-to-image generative model is a stochastic map $h \colon \mathcal{X} \to \mathcal{I}$ that takes a prompt $\mathbf{x}$ as input and generates an image $h(\mathbf{x}) \in \mathcal{I}$ that fits the description in the prompt $\mathbf{x}$. There are

---

[1]We will release the code and data upon paper publication.

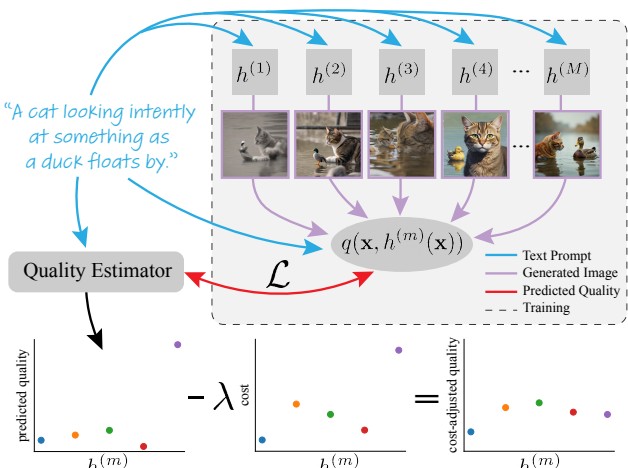

Figure 2: *Our pipeline.* During training (dashed box), a quality estimator is trained to predict per-prompt quality scores for all routing candidates $h^{(1)}, \ldots, h^{(M)}$. At inference time (bottom), given a prompt, predicted quality scores of all routing candidates are adjusted by their respective costs. The routing candidate that has the highest cost-adjusted score is chosen (see Eq. (3)).

many model classes one may use to construct such a model $h$, including conditional Generative Adversarial Networks (GANs) (Zhang et al., 2017; Goodfellow et al., 2014), Variational Auto-Encoder (VAE) (Kingma & Welling, 2022), and diffusion models (Ho et al., 2020), among others.

**Diffusion models** A specific class of text-to-image generative models that has recently been shown to produce high-fidelity images is given by diffusion-based models (Saharia et al., 2022; Ho et al., 2020; 2022). A diffusion generative model relies on a function $g \colon \mathcal{X} \times \mathbb{N} \times \mathbb{R}^D \to \mathcal{I}$ that takes as input a prompt $\mathbf{x}$, the number of denoising steps $T \in \mathbb{N}$, a noise vector $\mathbf{z} \in \mathbb{R}^D$ with $D = 3 \cdot WH$, and generates an image $\mathbf{i} = g(\mathbf{x}, T, \mathbf{z})$. Image generation is done by iteratively refining the initial noise vector $\mathbf{z}$ for $T$ iterations to produce the final image. The noise vector $\mathbf{z} \sim \mathcal{N}(\mathbf{0}, \mathbf{I})$ is typically sampled from the standard multivariate normal distribution and the $T$ refinement steps correspond to the reverse diffusion process, which reconstructs an image from a random initial state (Ho et al., 2020). With $\mathbf{z} \sim \mathcal{N}(\mathbf{0}, \mathbf{I})$ understood to be an implicit source of randomness, we define $h_T(\mathbf{x}) \doteq g(\mathbf{x}, T, \mathbf{z})$ to be an image sampled from the diffusion model using $T$ diffusion steps. With $T$ chosen, $h_T \colon \mathcal{X} \to \mathcal{I}$ is thus an instance of text-to-image generative models as described earlier. The importance of this view will be apparent when we describe our proposed method in Section 3, which enables an automatic selection of the number of denoising steps, separately for each prompt. Typically, the number of denoising steps is pre-chosen according to the computational budget available at inference time, with a low value of $T$, giving a lower computational cost at the expense of image quality.

## 3 Cost-Aware Text-To-Image Generation

We now describe our main proposal termed *CATImage* (Cost-Aware Text-based Image Generation), which seeks to minimize inference cost by adaptively adjusting the cost per prompt, depending on its complexity. As illustrated in Figure 1, in the case of a diffusion model, our key observation is that not all prompts require a large number of denoising steps to ensure quality. Thus, inference efficiency can be achieved by spending a small amount of computation for easy prompts. Our proposed framework is general and allows cost adjustment in a per-prompt manner via selecting an appropriate amount of resources from homogeneous choices (i.e., adaptively varying the number of denoising steps of a single diffusion model), or heterogeneous choices (i.e., adaptively route prompts to disparate, independent generative models).

We start by formalizing the cost-aware text-to-image generation task as a learning-to-route problem in Section 3.1. The formulation can be theoretically shown (Section 3.2) to have a simple Bayes optimal routing rule, involving subtracting off the expected quality metrics with the costs of candidate numbers of denoising steps. We show that the optimal rule can be estimated from data, and propose two estimators: a Transformer-based estimator (Vaswani et al., 2017), and a $K$-nearest neighbors (KNN) model.

### 3.1 Problem Formulation

Let $[n] \doteq \{1, 2, \ldots, n\}$ denote the set of counting numbers up to $n$. Suppose that we are given a fixed set of $M$ choices $\mathcal{H} \doteq \{h^{(1)}, \ldots, h^{(M)}\}$ where each choice $h^{(i)} \colon \mathcal{X} \to \mathcal{I}$ represents a trained generative model (see Section 2 for a precise definition). Our goal is to derive a routing rule that optimally (in the sense of quality-cost trade-offs) chooses the best model to invoke for each input prompt. These $M$ *base models* may be homogeneous, being derived from a single diffusion model with varying numbers of diffusion steps, a mix of heterogeneous generative model classes, or a combination of both. For example, if we want to decide whether to use 20 or 50 number of denoising steps in the Stable Diffusion XL (SDXL) model (Podell et al., 2023), then $M = 2$, and $\mathcal{H} = \{h^{(1)}, h^{(2)}\}$ where the two models are both SDXL with the number of denoising steps fixed to 20 and 50, respectively. We will abstract away the details of the underlying $M$ base models and propose a general framework that supports both the homogeneous and heterogeneous cases (as shown in our experiments in Section 5).

Suppose we are given a quality metric of interest $q \colon \mathcal{X} \times \mathcal{I} \to \mathbb{R}$ (see *Quality Metrics* under Section 5.1), which takes as input a prompt-image tuple, and estimates a quality score. We seek a router $r \colon \mathcal{X} \to [M]$ that predicts the index of the $M$ choices from a given prompt. We posit two desirable properties that the router ought to possess:

1. The router must respect a specified budget constraint on the inference cost.

2. Routing prompts to candidates in $\mathcal{H}$ must maximize average quality metric.

Following similar formulations considered in Jitkrittum et al. (2023; 2025); Mao et al. (2023); Tailor et al. (2024), the above desiderata may be realized as a constrained optimization problem:

$$\max_{r} Q(r) \quad \text{subject to} \quad C(r) \leq B, \quad \text{where} \tag{1}$$

$$Q(r) \doteq \mathbb{E}\left[ \sum_{m \in [M]} \mathbf{1}\left[r(\mathbf{x}) = m\right] \cdot q(\mathbf{x}, h^{(m)}(\mathbf{x})) \right], \quad \text{and} \quad C(r) \doteq \mathbb{E}\left[ \sum_{m \in [M]} \mathbf{1}\left[r(\mathbf{x}) = m\right] \cdot c^{(m)} \right], \tag{2}$$

where for $m \in [M]$, $c^{(m)} \geq 0$ denotes the cost for the model $h^{(m)}$ to produce one image for a given prompt, $\mathbb{E}$ denotes the expectation with respect to the population joint distribution on all random variables (i.e., prompt $\mathbf{x}$, and the sampled output of $h^{(m)}$), $B \geq 0$ is a hyperparameter specifying an upper bound on the average cost. The optimization problem (1) thus seeks a router $r$ that maximizes the average quality $Q(r)$ subject to the constraint that the average cost (over all prompts) is bounded above by $B$.

*Remark.* The optimization problem is general and allows the per-model costs to be in any unit suitable for the application (e.g., latency in seconds, FLOP counts). Further, no practical constraint is imposed on the quality metric function $q$. For instance, $q$ could be the CLIP score (Radford et al., 2021). Intuitively, if the budget $B$ is large, the cost constraint $C(r) \leq B$ would have little effect, and the optimal router is expected to route each prompt to the base model that can produce the highest quality metric score, disregarding the cost of the model. In practice, such a model is often the largest one in the pool $\mathcal{H}$, or the diffusion model with the largest number of denoising steps. On the contrary, if $B$ is small, the router would prioritize cost over quality, preferring to choose a small base model (or a small number of denoising steps) over a larger candidate. This proposal offers a framework to allow trading off average quality with cost in a unified way by varying $B$.

### 3.2 Theoretically Optimal Routing Rule

Having formulated the constrained problem in (1), we now investigate its theoretically optimal solution. We will use the optimal solution to guide us on how to design a practical router. Based on the results in Jitkrittum et al. (2023; 2025), the optimal solution to (1) is shown in Proposition 1.

**Proposition 1.** *For a cost budget $B > 0$, the optimal router $r^* \colon \mathcal{X} \to \{1, \ldots, M\}$ to the constrained optimization problem (1) is*

$$r^*(\mathbf{x}) = \arg \max_{m \in [M]} \mathbb{E}\left[q(\mathbf{x}, h^{(m)}(\mathbf{x})) \mid \mathbf{x}\right] - \lambda \cdot c^{(m)},$$

*where the conditional expectation is over the sampled output from the model $h^{(m)}$, and $\lambda \geq 0$ is a Lagrange multiplier inversely proportional to $B$.*

The result follows from Proposition 1 in Jitkrittum et al. (2025). The result states that the choice/model we choose to route a prompt $\mathbf{x}$ to is the one that maximizes the average quality, adjusted additively by the cost of the model. The hyperparameter $\lambda$ controls the trade-off between quality and cost, and is inversely proportional to the budget $B$. For instance, if $\lambda = 0$ (corresponding to $B = \infty$), then the model with the highest expected quality for $\mathbf{x}$ will be chosen, regardless of its cost. Increasing $\lambda$ enforces the routing rule to account more for model costs, in addition to the expected quality.

**Estimating the Optimal Rule**    The optimal rule $r^*$ in Proposition 1 depends on the population conditional expectation $\gamma^{(m)}(\mathbf{x}) \doteq \mathbb{E}\left[q(\mathbf{x}, h^{(m)}(\mathbf{x})) \mid \mathbf{x}\right]$, which is unknown. Following a similar reasoning as in Jitkrittum et al. (2025), we propose plugging in an empirical estimator $\hat{\gamma}^{(m)} \colon \mathcal{X} \to \mathbb{R}$ in place of $\gamma^{(m)}$, resulting in the empirical rule $\hat{r}_\lambda$:

$$\hat{r}_\lambda(\mathbf{x}) = \arg \max_{m \in [M]} \hat{\gamma}^{(m)}(\mathbf{x}) - \lambda \cdot c^{(m)}. \tag{3}$$

For each $m \in [M]$, the idea is to train an estimator $\hat{\gamma}^{(m)}$ to estimate the true expected quality. That is, suppose we are given a collection of $N$ training prompts $\{\mathbf{x}_i\}_{i=1}^N$. For each prompt $\mathbf{x}_i$, we may sample $S$ times from $h^{(m)}$ to produce output images $\mathbf{i}_{i,1}^{(m)} \ldots, \mathbf{i}_{i,S}^{(m)}$. These output images allow one to estimate the empirical expectation of the quality $\hat{y}_i \doteq \frac{1}{S} \sum_{s=1}^S q(\mathbf{x}, \mathbf{i}_{i,s}^{(m)})$. With the labeled training set $\{(\mathbf{x}_i, \hat{y}_i)\}_{i=1}^N$, we may then proceed to train a predictive model $\hat{\gamma}(\mathbf{x}) \doteq \left(\hat{\gamma}^{(1)}(\mathbf{x}), \ldots, \hat{\gamma}^{(M)}(\mathbf{x})\right)$, which has $M$ output heads for predicting the expected qualities of the $M$ models. There are several standard machine learning models one can use as the model class for $\hat{\gamma}$.

We emphasize that we do not advocate a specific model class as part of our proposal since different model classes offer distinct properties on training and inference costs, which may be best tailored to the application. What we propose is an application of the generic routing rule in (3) to text-to-image model routing. The rule is guaranteed to give a good quality-cost trade-off provided that the estimator $\hat{\gamma}^{(m)}$ well estimates $\gamma^{(m)}$. In experiments (Section 5), we demonstrate estimating $\gamma^{(m)}$ with two model classes: 1) $K$-nearest neighbors, and 2) Multi-Layer Perceptron (MLP) with a Transformer backbone (Vaswani et al., 2017). Likewise, we do not propose or advocate a specific value of $\lambda$. The parameter is left to the user as a knob to control the desired degree of quality-cost trade-off. In experiments, we evaluate our proposed routing rule by considering a wide range of $\lambda$ and show the trade-off as a deferral curve (see Section 3.3). An illustration summarizing our pipeline is displayed in Figure 2.

### 3.3    Deferral Curve

In general, any methods that offer the ability to trade off quality and cost may be evaluated via a *deferral curve* (Bolukbasi et al., 2017; Cortes et al., 2016; Gupta et al., 2024; Narasimhan et al., 2022). A deferral curve is a curve showing the average quality against the average cost, in a quality-cost two-dimensional plane. Specifically, for our proposed routing rule $\hat{r}_\lambda$ in (3), the curve is precisely given by $\mathcal{C} = \{(C(\hat{r}_\lambda), Q(\hat{r}_\lambda)) \mid \lambda \in [0, \infty)\}$ where $Q$ and $C$ denote the average quality and cost, and are defined in Eq. (2). In practice, the population expectation in $Q$ and $C$ is replaced with an empirical expectation over examples in a test set. We generate the deferral curve of our routing decision by sweeping $\lambda$ from 0 to a sufficiently large value. This allows the router to transition from maximizing quality at $\lambda = 0$ to selecting the cheaper candidates as $\lambda$ increases. More generally, one evaluates the deferral curve of a method by computing its average quality and cost as we vary parameters that control the trade-off. For instance, for the SDXL diffusion model, we may produce a deferral curve by varying the number of denoising steps.

## 4    Related Work

**Uniform Optimization Strategies for Diffusion Models**    Diffusion models have recently exploded in popularity due to their high performance on tasks such as image and video generation, audio generation,

and 3D shape generation (Ho et al., 2020; Ramesh et al., 2021). Latent diffusion models (Rombach et al., 2022) have significantly improved training and inference efficiency, but still require a large number of forward denoising neural network evaluations to produce high-quality results. To address this, an extensive body of literature has been proposed to optimize and accelerate diffusion models, which are typically applied *uniformly across all prompts.* For example, optimizing the sampling strategy may enable more efficient denoising computation (Li et al., 2024; Chen et al., 2023b; Li et al., 2023a), such as timestep integration (Nichol & Dhariwal, 2021) or conditioning on the denoising (Preechakul et al., 2022). Optimizing solvers for the denoising step can also efficiently reduce the computation to avoid re-training or fine-tuning (Song et al., 2020; Lu et al., 2022; Liu et al., 2022; Karras et al., 2022). Alternatively, reducing the redundant computations by caching the internal results within the denoising network is also explored in (Ma et al., 2024a;b). Another common approach includes model-based optimizations, such as distilling a fully trained model into a smaller student model that achieves comparable results with fewer denoising steps (Sauer et al., 2024; Salimans & Ho, 2022; Meng et al., 2023; Liu et al., 2023) or combining multiple denoising models with different sizes to accelerate the denoising process (Yang et al., 2023; Li et al., 2023b; Pan et al., 2024). An alternative strategy is to approximate the direct mapping from initial noise to generated images, further reducing the number of denoising steps (Luo et al., 2023; Song et al., 2023).

**Adaptive Optimization Strategies for Diffusion Models**  Instead of a fixed reduction in computational resources, AdaDiff (Tang et al., 2023) explores a more dynamic approach where the number of denoising steps is decided based on the uncertainty estimation of the intermediate results during denoising. Our work shares a similar motivation for flexible resource allocation. However, we adaptively allocate resources according to prompt complexity and thus can select the most suitable number of steps or model before any denoising process. Concurrently, AdaDiff (Zhang et al., 2023) tackles optimal number of steps selection using a prompt-specific policy, with a lightweight network trained on a reward function that balances image quality and computational resources. In contrast, we decouple the quality estimation from the routing decision, which allows our framework to adapt to different resource constraints without any retraining.

**Learning-To-Defer, and Modeling Routing**  The idea of adaptively invoking a different expert on each input is a widely studied area in machine learning under the topic of *learning to defer.* Here, each expert may be a human expert (Mozannar & Sontag, 2020; Mozannar et al., 2023; Sangalli et al., 2023), or a larger model (Narasimhan et al., 2022; Jitkrittum et al., 2023; Mao et al., 2023; Gupta et al., 2024). In the latter, depending on the topology or order the models are invoked, a learning-to-defer method may yield a *cascade* if models are arranged in a chain (Wang et al., 2022a; Jitkrittum et al., 2023; Kolawole et al., 2024); or yield a *routed model* if there is a central routing logic (i.e., the router) which selectively sends input traffic to appropriate models (Jiang et al., 2023; Mao et al., 2023; Gupta et al., 2024; Jitkrittum et al., 2025). The latter setup is also known as *model routing* and receives much attention of late, especially in the natural language processing literature. Model routing has been successfully applied to route between many Large Language Model (LLMs) of various sizes and specialties (see Chen et al. (2023a); Hu et al. (2024); Zhuang et al. (2025); Ong et al. (2025); Jitkrittum et al. (2025) and references therein). To our knowledge, our work is one of the first that connects the model routing problem to efficient text-to-image generation.

## 5 Experiments

In this section, we show how our proposed routing method (Section 3) can be realized in practice by evaluating its effectiveness on real data. We experiment with both homogeneous (i.e., all routing candidates are derived from the same diffusion model with different candidate numbers of denoising steps), and heterogeneous settings (i.e., the routing candidates also include different generative models). Our goal is to optimally select the best model (or number of denoising steps) for each input prompt given a specified cost constraint.

### 5.1 Experimental Setup

**Text-To-Image Generative Models**  As defined in Section 3.1, our method selects from a set of generative models $\mathcal{H}$ for each input prompt. We consider a diverse range of models with varying configurations, each offering a different trade-off between image quality and computational cost:

1. SDXL: a widely-used SD architecture (Rombach et al., 2022). To see the full extent of the achievable trade-off, we consider representative numbers of denoising steps in a wide range between 1 and 100.

2. TURBO (Sauer et al., 2024) and LIGHTNING (Lin et al., 2024): distilled versions of SDXL for faster generation. We use the SDXL variant with 1 step for Turbo, and 4 steps for Lighting.

3. DDIM (Song et al., 2020): a non-Markovian diffusion process allowing faster sampling. We use this sampling strategy on the SDXL variant at 50 steps.

4. DEEPCACHE (Ma et al., 2024b): a caching method that reduces redundant computation in SDXL. We use the official implementation released from Ma et al. (2024b), and set the cache interval parameter to 3.

5. INFINITY (Han et al., 2024): a non-diffusion, auto-regressive text-to-image model based on the Transformer encoder-decoder. We use the pre-trained Infinity-2B variant with a visual vocabulary size of $2^{32}$.

**Quality Metrics** The effectiveness of generative models largely depends on the criteria used to evaluate their output. Our proposed method can adaptively identify the optimal allocation of generative model for *any* instance-level image quality metric. As there is no consensus on the optimal metric for evaluating image quality, we explore several widely-used metrics: *CLIPScore* (Radford et al., 2021) for text-image semantic alignment, *ImageReward* (Xu et al., 2023) with a reward model tuned to human preferences, and *Aesthetic Score* (Beaumont & Schuhmann, 2022) trained on human aesthetic ratings from LAION (Schuhmann et al., 2022). Additionally, we also introduce *Sharpness* metric adapted from Paris et al. (2011), defined as, $q_{\text{Sharp}}(\mathbf{x}, \mathbf{i}) = \frac{\sum_{ij}(\mathbf{i}_{ij} - [\mathbf{i} \circledast G]_{ij})^2}{\sum_{ij} \mathbf{i}_{ij}^2}$, where $\circledast$ denotes the convolution operator, $\mathbf{i}_{i,j}$ is the pixel intensity at location $(i, j)$, and $G$ is a Gaussian kernel with standard deviation of 1. Intuitively, this metric measures the relative distance between the given image $\mathbf{i}$ and itself after a Gaussian blur filter is applied.

**Quality Estimator $\hat{\gamma}$** One of the key components of our routing method is the quality estimator which estimates the expected quality of the $m$-th model given an input prompt (see $\hat{\gamma}^m$ in Eq. (3)). We explore two model classes: a K-NEAREST NEIGHBORS ($K$-NN) model and a TRANSFORMER-based model. Both of these models incur a negligible inference cost: less than 0.001 TFLOPs compared to 1.5 TFLOPs of the smallest base model in the pool (Infinity).

The $K$-NN approach provides a non-parametric way to estimate quality by averaging the quality scores of $K$ nearest training prompts in the space of CLIP embeddings (Radford et al., 2021). This method is simple, and can generalize well with sufficient data. The Transformer model takes as input the per-token embeddings produced by the frozen CLIP text encoder. A two-layer MLP with $M$ output heads is added to each output token embedding. Pooling across all tokens gives $M$ output scores $\hat{\gamma}^{(1)}(\mathbf{x}), \dots, \hat{\gamma}^{(M)}(\mathbf{x})$ (see Eq. (3)), each estimating the expected quality of the $m$-th model on prompt $\mathbf{x}$ (see Appendix Section D for details).

All base models except Infinity already use CLIP embeddings, making router overhead negligible. Infinity uses Flan-T5 embeddings ($\approx 13$ GFLOPs overhead), but this cost is minimal compared to one SDXL call ($\approx 200$ TFLOPs for 17 steps).

We train a separate model for each of the quality metrics considered. In each case, the quality scores are linearly scaled across all training examples to be in [0, 1]. These scaled metrics are treated as ground-truth probabilities, and the model is trained by minimizing the sum of the sigmoid cross-entropy losses across all heads.

## 5.2 Dataset Details

We utilize two datasets: 1) the COCO captioning dataset (Lin et al., 2014), which contains high-quality and detailed image captioning, and 2) the DiffusionDB dataset (Wang et al., 2022b), which contains a larger collection of realistic, user-generated text prompts for text-to-image generation. From both datasets, we sub-sample prompts by retaining only those with pairwise CLIP similarity below 0.75, resulting in a diverse set of 18,384 prompts in COCO dataset, and 97,841 prompts on the DiffusionDB dataset. We split each dataset independently into 80% for training, 10% for validation, and 10% for testing. We then generate

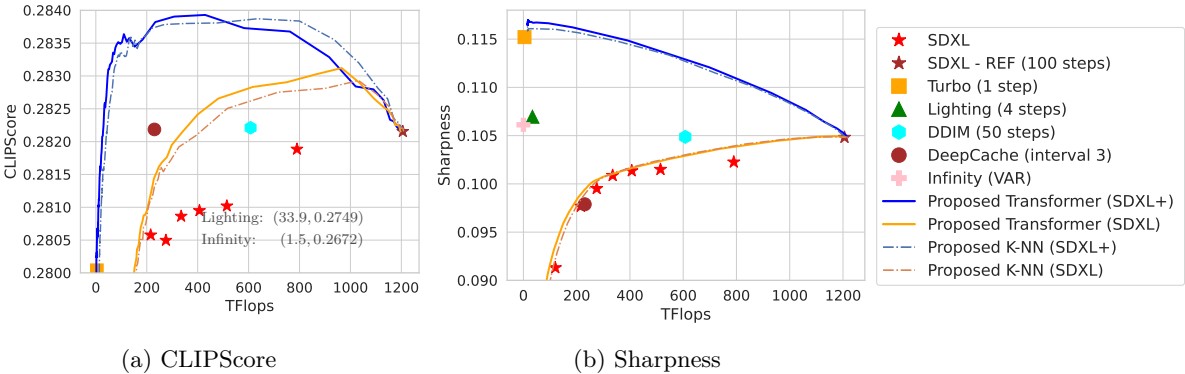

(a) CLIPScore  (b) Sharpness

Figure 3: Deferral curves of our proposed methods and baselines on the COCO dataset. Each data point represents the average quality and cost across the entire test set (1839 prompts). Quality is measured by CLIPScore in (a) and pixel sharpness in (b), as defined in Section 5.1. Our Proposed Transformer (SDXL+), which selects from all available SDXL denoising steps and baseline models, offers the best quality-cost trade-off (TFLOPs). Baselines that are not visible are shown at the bottom-right corner in the format of (cost, CLIPScore).

images from those prompts using all the base text-to-image models as described earlier. For SDXL, we generate images with various numbers of denoising steps ranging from 1 to 100. The costs in terms of FLOPs from these candidates cover the full range of costs of all other baselines.

For each model, we generate four images per prompt (i.e., $S = 4$ in Section 3.2) using different random seeds, with a fixed seed across different numbers of steps for SDXL. The generated images for each prompt $\mathbf{x}_i$ allow us to compute the average quality metric, which is then used as the training label $\hat{y}_i$ (as described in Section 3.2). Unless otherwise specified, we use the widely used Euler Scheduler (Karras et al., 2022) for diffusion-based image generation.

## 5.3 Experiments on COCO dataset

We present experimental results on a subset of COCO's test set (Lin et al., 2014) consisting of 1.8k image-caption pairs in Figure 3. We evaluate the deferral curves (see Section 3.3) of our proposed method and all the baselines. The results are shown in Figures 3a and 3b for the two different quality metrics: CLIPScore and image sharpness (Section 5.1), respectively. The deferral curves plot average quality against average cost measured in TFLOPs (Tera Floating Point Operations). Baselines that do not support dynamic quality-cost trade-off are shown as isolated dots in the same quality-cost plane; these baselines use the same compute cost for image generation for each input prompt. For instance, each point ★ of SDXL represents the performance of the SDXL model with the number of denoising steps fixed. For our proposed methods, *Proposed (SDXL)* refers to the homogeneous configuration in which the model candidate set $\mathcal{H}$ consists solely of the SDXL model at multiple numbers of denoising steps settings. *Proposed (SDXL+)* extends this configuration by incorporating other text-to-image models considered, namely, Turbo, DDIM, DeepCache, and Infinity. Each of these has two variants based on Transformer or $K$-NN as the model class for estimating the expected quality metric.

**Homogeneous vs. Heterogeneous setting** In both settings, our methods outperform baselines with static inference costs per prompt. The heterogeneous setting further benefits from models with strong quality-to-cost trade-offs (e.g., Infinity, Turbo), improving our dynamic routing's effectiveness and cost-efficiency. Moreover, our strategy remains adaptive, seamlessly allocating prompts to higher-performance models when additional computational resources are available, improving performance beyond what is attainable using each model alone (see Appendix Section G for details on model selection rates).

Table 1: We train the k-NN quality estimator over $T = 100$ trials on COCO. In each trial, we generate $S$ images for each training prompt with 1) $S = 1$ and 2) $S = 3$. Each of these two cases results in $T$ k-NN models, and hence $T$ (random) deferral curves evaluated on the same test set as used in Fig 3. We report the mean performance (CLIPScore) across the $T$ trials for our approach.

| **CLIPScore** $\uparrow$ | | | | | | | | |
|---|---|---|---|---|---|---|---|---|
| | Infinity | Turbo | Lighting | SDXL-9 | DeepCache | SDXL-22 | DDIM | SDXL-65 |
| Ours (S=1) | 0.2816 | 0.2814 | 0.2828 | 0.2828 | 0.2828 | 0.2828 | 0.2824 | 0.2819 |
| Ours (S=3) | 0.2816 | 0.2816 | 0.2831 | **0.2832** | **0.2832** | **0.2832** | 0.2828 | 0.2819 |
| Fixed | 0.2816 | 0.2794 | 0.2798 | 0.2773 | 0.2749 | 0.2805 | 0.2820 | 0.2819 |
| Win Rate (S=1) | - | 1 | 1 | 1 | 1 | 1 | 0.98 | - |
| Win Rate (S=3) | - | 1 | 1 | 1 | 1 | 1 | 1 | - |
| TFLOPs | 1.5 | 1.54 | 23.92 | 107.73 | 210 | 263.34 | 598.5 | 778.05 |

**Transformer vs. KNN** Between the two proposed variants, the Transformer-based variant generally outperforms the $K$-NN variant, suggesting that directly learning to predict the quality metric can be more effective than estimating it from neighboring prompts.

## 5.4 Effect of Sample Size S

As the estimation error depends on the randomness in image generation, we aggregate signals from multiple generated images by averaging their quality metric values across multiple random seeds. To further quantify variability across trials, we vary the number of generated images for each prompt and train a separate model for each case. In Table 1, we report the mean performance of CLIPScores on the COCO dataset, as well as the Win Rate, defined as the fraction of trials that our router has higher average quality across the test set than the baseline. The maximum standard deviation across trials of both above variants is less than $2 \times 10^{-4}$ throughout the cost range.

The results show that using S=3 improves the test performance compared to S=1. Additionally, the win rate of "Ours (S=1)" compared to the baseline is 100% in almost all cost ranges. This win rate implies that our approaches are statistically significantly better than the baseline according to the sign test at significance level $a < 10^{-6}$. This means using one image per prompt is already sufficient to improve the baseline of using fixed compute costs. Deviation across trials is minimal relative to the mean metric, suggesting that S=3 is sufficient. In all other experiments in the paper, we used S=4.

## 5.5 Experiments on DiffusionDB dataset

In this section, we present results on a subset of prompts from the DiffusionDB dataset (Wang et al., 2022b), which aligns more closely with real-world prompts used in text-to-image generation. We evaluate the performance across four metrics: *CLIPScore*, *ImageReward*, *Aesthetic Score*, and *Sharpness*.

Quantitative results comparing our dynamic routing method to the fixed-model baselines are summarized in Table 2. This table effectively captures the trade-offs shown in the deferral curves at a specific cost equal to each baseline. We use KNN as a quality estimator to efficiently evaluate multiple metrics at scale. For reference, we also provide the Oracle performance, which selects the optimal candidate for each prompt based on ground-truth quality scores rather than predicted estimates. This is an absolute upper bound on the best quality-cost trade-off attainable by any routing methods. In practice, it is extremely challenging to realize a quality-cost operating point that is close to that of the Oracle (see Hu et al. (2024)). The results show that our method consistently matches or exceeds fixed-model baseline performance across all four quality metrics. Additionally, the highest value of each score (highlighted in Table 2 in bold) is attainable *only* with our routing strategy. In other words, even under an unconstrainedand computational budget, none of

Table 2: Quality-cost trade-off of our proposed approach on DiffusionDB (Section 5.5). We report the average quality achieved by our routing approach when operating at the cost (TFLOPs) of each model in the pool. For each metric, the highest score achieved between our approach and the fixed baseline is highlighted in bold. The Oracle performance is provided in gray for context.

**CLIPScore** (Radford et al., 2021) ↑

| | | | | | | | | | |
|---|---|---|---|---|---|---|---|---|---|
| Oracle | $0.259_{\pm 6\text{e-}4}$ | $0.309_{\pm 4\text{e-}4}$ | $0.323_{\pm 4\text{e-}4}$ | $0.334_{\pm 4\text{e-}4}$ | $0.337_{\pm 4\text{e-}4}$ | $0.337_{\pm 4\text{e-}4}$ | $0.336_{\pm 4\text{e-}4}$ | $0.336_{\pm 4\text{e-}4}$ | $0.318_{\pm 4\text{e-}4}$ |
| Ours | $0.259_{\pm 6\text{e-}4}$ | $0.304_{\pm 4\text{e-}4}$ | $0.308_{\pm 4\text{e-}4}$ | $0.314_{\pm 4\text{e-}4}$ | $0.316_{\pm 4\text{e-}4}$ | $0.317_{\pm 4\text{e-}4}$ | $\mathbf{0.318}_{\pm 4\text{e-}4}$ | $0.318_{\pm 4\text{e-}4}$ | $0.318_{\pm 4\text{e-}4}$ |
| Fixed | $0.259_{\pm 6\text{e-}4}$ | $0.304_{\pm 4\text{e-}4}$ | $0.300_{\pm 4\text{e-}4}$ | $0.308_{\pm 4\text{e-}4}$ | $0.316_{\pm 4\text{e-}4}$ | $0.315_{\pm 4\text{e-}4}$ | $0.317_{\pm 4\text{e-}4}$ | $0.315_{\pm 4\text{e-}4}$ | $0.318_{\pm 4\text{e-}4}$ |

**Sharpness** (Section 5.1) ↑

| | | | | | | | | | |
|---|---|---|---|---|---|---|---|---|---|
| Oracle | $0.131_{\pm 4\text{e-}4}$ | $0.140_{\pm 3\text{e-}4}$ | $0.144_{\pm 3\text{e-}4}$ | $0.145_{\pm 3\text{e-}4}$ | $0.146_{\pm 3\text{e-}4}$ | $0.146_{\pm 3\text{e-}4}$ | $0.145_{\pm 4\text{e-}4}$ | $0.145_{\pm 4\text{e-}4}$ | $0.126_{\pm 3\text{e-}4}$ |
| Ours | $0.131_{\pm 4\text{e-}4}$ | $0.135_{\pm 3\text{e-}4}$ | $0.135_{\pm 3\text{e-}4}$ | $0.136_{\pm 3\text{e-}4}$ | $\mathbf{0.137}_{\pm 3\text{e-}4}$ | $0.137_{\pm 3\text{e-}4}$ | $0.137_{\pm 3\text{e-}4}$ | $0.137_{\pm 3\text{e-}4}$ | $0.126_{\pm 3\text{e-}4}$ |
| Fixed | $0.131_{\pm 4\text{e-}4}$ | $0.122_{\pm 3\text{e-}4}$ | $0.110_{\pm 3\text{e-}4}$ | $0.103_{\pm 2\text{e-}4}$ | $0.101_{\pm 2\text{e-}4}$ | $0.114_{\pm 2\text{e-}4}$ | $0.123_{\pm 3\text{e-}4}$ | $0.107_{\pm 3\text{e-}4}$ | $0.126_{\pm 3\text{e-}4}$ |

**Aesthetic Score** (Beaumont & Schuhmann, 2022) ↑

| | | | | | | | | | |
|---|---|---|---|---|---|---|---|---|---|
| Oracle | $6.824_{\pm 8\text{e-}3}$ | $7.132_{\pm 8\text{e-}3}$ | $7.385_{\pm 8\text{e-}3}$ | $7.426_{\pm 7\text{e-}3}$ | $7.421_{\pm 8\text{e-}3}$ | $7.416_{\pm 8\text{e-}3}$ | $7.284_{\pm 8\text{e-}3}$ | $7.284_{\pm 8\text{e-}3}$ | $6.707_{\pm 8\text{e-}3}$ |
| Ours | $6.824_{\pm 8\text{e-}3}$ | $6.913_{\pm 9\text{e-}3}$ | $\mathbf{7.042}_{\pm 9\text{e-}3}$ | $7.032_{\pm 9\text{e-}3}$ | $7.012_{\pm 9\text{e-}3}$ | $7.012_{\pm 9\text{e-}3}$ | $6.935_{\pm 9\text{e-}3}$ | $6.935_{\pm 9\text{e-}3}$ | $6.707_{\pm 8\text{e-}3}$ |
| Fixed | $6.824_{\pm 8\text{e-}3}$ | $6.780_{\pm 9\text{e-}3}$ | $7.010_{\pm 9\text{e-}3}$ | $6.285_{\pm 8\text{e-}3}$ | $6.625_{\pm 9\text{e-}3}$ | $6.588_{\pm 8\text{e-}3}$ | $6.690_{\pm 8\text{e-}3}$ | $6.600_{\pm 9\text{e-}3}$ | $6.707_{\pm 8\text{e-}3}$ |

**ImageReward** (Xu et al., 2023) ↑

| | | | | | | | | | |
|---|---|---|---|---|---|---|---|---|---|
| Oracle | $1.029_{\pm 9\text{e-}3}$ | $1.303_{\pm 6\text{e-}3}$ | $1.416_{\pm 5\text{e-}3}$ | $1.446_{\pm 5\text{e-}3}$ | $1.446_{\pm 5\text{e-}3}$ | $1.444_{\pm 5\text{e-}3}$ | $1.376_{\pm 5\text{e-}3}$ | $1.376_{\pm 5\text{e-}3}$ | $0.891_{\pm 7\text{e-}3}$ |
| Ours | $1.029_{\pm 9\text{e-}3}$ | $1.083_{\pm 8\text{e-}3}$ | $\mathbf{1.086}_{\pm 8\text{e-}3}$ | $1.086_{\pm 8\text{e-}3}$ | $1.079_{\pm 8\text{e-}3}$ | $1.076_{\pm 8\text{e-}3}$ | $1.037_{\pm 8\text{e-}3}$ | $1.037_{\pm 8\text{e-}3}$ | $0.891_{\pm 7\text{e-}3}$ |
| Fixed | $1.029_{\pm 9\text{e-}3}$ | $0.960_{\pm 8\text{e-}3}$ | $0.932_{\pm 8\text{e-}3}$ | $0.497_{\pm 8\text{e-}3}$ | $0.809_{\pm 9\text{e-}3}$ | $0.769_{\pm 8\text{e-}3}$ | $0.866_{\pm 8\text{e-}3}$ | $0.861_{\pm 9\text{e-}3}$ | $0.891_{\pm 7\text{e-}3}$ |

| | INFI | TURB | LIGH | SDXL | DEEP | SDXL | SDXL | DDIM | SDXL |
|---|---|---|---|---|---|---|---|---|---|
| **TFLOPs** | **1.50** | **1.54** | **23.92** | **119.70** | **210.00** | **239.40** | **598.50** | **598.50** | **1197.00** |

the individual baselines can attain the quality that our adaptive routing achieves through prompt-based allocation across the model pool.

Table 3 quantifies the computational cost reduction achieved by our routing method compared to the baseline at equivalent quality levels (on Sharpness metric). For inherently efficient models (e.g. Infinity(Han et al., 2024), Turbo (Sauer et al., 2024)), the savings appear marginal. However, compared to Lighting (Lin et al., 2024), a *distilled* SDXL variant, our method achieves the same performance at only 6% of its computational cost. For higher-performance models, such as SDXL at 100 denoising steps, the savings are even more significant.

Table 3: Cost ratio (%) of our method compared to baselines to match the quality score (Sharpness)

| Model | Our cost |
|---|---|
| Infinity | 100% |
| Turbo | 97.40% |
| Lighting | 6.27% |
| DeepCache | 0.71% |
| DDIM | 0.25% |
| SDXL100 | 0.13% |

## 5.6 Incorporating FLUX.1-dev model

The quality-cost trade-off of our router improves as we add better models to the pool. With the same experimental setup on DiffusionDB as in Section 5.5, we now add a high-performing model FLUX.1-dev[2] at three different denoising steps (1, 15, and 30), resulting in a total of 12 models in the routing pool. The addition of a new model results in an improved performance across all cost ranges compared to using any individual model alone, as shown in Table 4. Specifically, at an average cost of less than 600 TFLOPs, our router achieves an aesthetic score of 7.109, surpassing the best single model (FLUX-30) which scores 6.98 at a fixed cost of 1785.6 TFLOPs. The router learns that the most powerful model (e.g., FLUX-30) is not necessarily the best for every input; a cheaper model can sometimes yield superior results on prompts for which it is better suited.

## 5.7 Qualitative Comparison

To better illustrate the effects of routing, we further present qualitative comparisons between our routing results and the FLUX model (with 30 denoising steps) in Figure 4. We perform routing using Aesthetic

---

[2]https://huggingface.co/black-forest-labs/FLUX.1-dev

Table 4: Average quality achieved by our routing approach when operating at the cost (TFLOPs) of each model in the pool. In this experiment, we include FLUX.1-dev at 1, 15, and 30 denoising steps, for the total of 12 models in the candidate pool.

| Aesthetic Score ↑ | | | | | | | | | | | |
|---|---|---|---|---|---|---|---|---|---|---|---|
| | Infinity | Turbo | Lighting | FLUX-1 | SDXL-10 | DeepCache | SDXL-20 | DDIM | SDXL-50 | FLUX-15 | SDXL100 | FLUX-30 |
| Ours | 6.824 | 6.913 | 7.045 | 7.055 | 7.068 | 7.082 | 7.087 | **7.109** | 7.109 | 7.108 | 7.090 | 6.983 |
| Fixed | 6.824 | 6.780 | 7.010 | 3.160 | 6.285 | 6.625 | 6.588 | 6.600 | 6.690 | 6.952 | 6.707 | 6.983 |
| TFLOPs | 1.50 | 1.54 | 23.92 | 59.52 | 119.70 | 210 | 239.4 | 598.5 | 598.5 | 892.8 | 1197.00 | 1785.60 |
| Fraction of TFLOPs | 0.1% | 0.1% | 1.3% | 3.3% | 6.7% | 11.8% | 13.4% | 33.5% | 33.5% | 50.0% | 67.0% | 100.0% |

Score, which has been shown to correlate well with human preference (Section A). Note that in this case, our routing framework can be used to select the number of denoising steps (i.e. the same FLUX model with 15 denoising steps) or to select an entirely different model (Infinity, Turbo, etc.). Interestingly, we observe that the router tends to select *Infinity* for scenic prompts and *Turbo* for stylized (cartoon/painting) images, whereas for general/human related prompts it typically reduces the number of denoising steps from 30 to 15.

## 6 Conclusion and Future Work

In this paper, we present *CATImage*, a cost-aware routing approach that dynamically selects optimal models and numbers of denoising steps based on prompt complexity. We show that incorporating multiple base models, such as distilled versions of diffusion models and alternative architectures, improves the quality–cost trade-off. Extensive experiments on COCO and DiffusionDB datasets across multiple quality metrics validate our method's effectiveness and generalization capability. Nevertheless, several limitations are worth highlighting. To determine optimal routing decisions, *CATImage* relies on estimating the expected quality *per* prompt, which excludes metrics such as Fréchet Inception Distance (FID) (Heusel et al., 2017) that measure statistical similarity across the entire image distribution. Additionally, the routing problem we consider assumes a static pool setup where the set of text-to-image models is considered fixed. In practice, the model pool can change which can add additional maintenance cost of retraining the router. Addressing these limitations remains a direction for future research.

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

**Prompt:** an epic painting of a futuristic solitary astronaut walking along the icy surface of Titan, Saturn is visible through the haze, unreal 5, DAZ, detailed, soft focus, brilliant, 4k, 8k, HD, trending on artstation, art by Rick Guidice painting by Robert McCall by John Harris, abstract

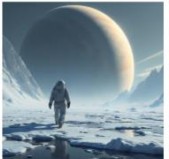 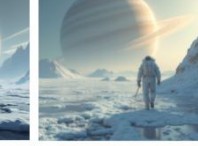

**Router's Choice (Infinity) Save 99%**     **Fixed Cost (FLUX-30))**

**Prompt:** Looking towards The Ettenmoors from The Mountains of Angmar. The ruin of Weathertop can be seen in the far distance on the horizon, Eriador, Middle Earth, Tolkien, by Albert Bierstadt and Henri Mauperch

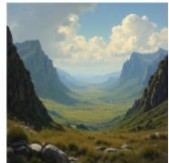 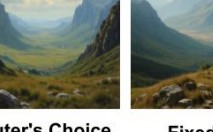

**Router's Choice (Infinity) Save 99%**     **Fixed Cost (FLUX-30))**

**Prompt:** awe inspiring bruce pennington landscape, orange sky, red cyan forest digital art painting of 1960 s, japan at night, 4k, 8k, hyperdetailed, minimalist

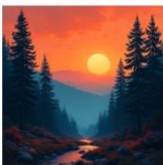 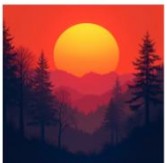

**Router's Choice (Infinity) Save 99%**     **Fixed Cost (FLUX-30))**

**Prompt:** excited brown and tan domino finnish lapphund, studio ghibli art style

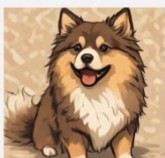 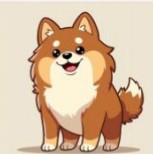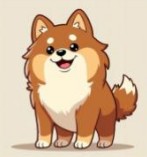

**Router's Choice (Turbo) Save 99%**     **Fixed Cost (FLUX-30))**

**Prompt:** A tabby cat wearing clothes designed by Issey Miyake painted by Alex Katz

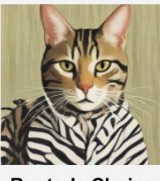 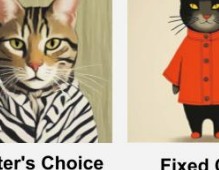

**Router's Choice (Turbo) Save 99%**     **Fixed Cost (FLUX-30))**

**Prompt:** portrait of elton john lennon thoughtful in 1970 by ilya repin

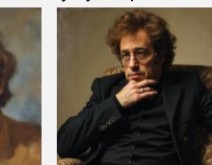

**Router's Choice (Turbo) Save 99%**     **Fixed Cost (FLUX-30))**

**Prompt:** photograph of Kim Kardashian cutting a banana, with a pen, in a tent

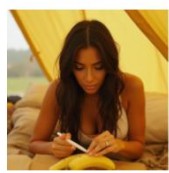 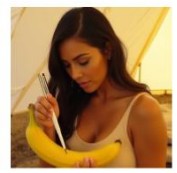

**Router's Choice (FLUX-15) Save 50%**     **Fixed Cost (FLUX-30))**

**Prompt:** 1930's photo of cat on a rooftop in japan

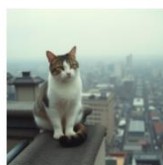 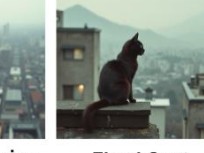

**Router's Choice (FLUX-15) Save 50%**     **Fixed Cost (FLUX-30))**

**Prompt:** cinematic scene with mckenna grace as eowyn from lord of the rings, live action film, battle armor chain mail, dramatic, small details, volumetric lighting, still frame

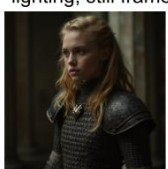 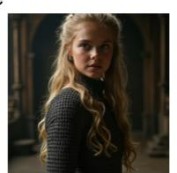

**Router's Choice (FLUX-15) Save 50%**     **Fixed Cost (FLUX-30))**

Figure 4: Example results from our routing framework compared to a fixed FLUX model (30 denoising steps). Examples are grouped by the three most frequently selected models: Infinity, Turbo, and FLUX-15. We observe a tendency for the router to select Infinity for scenic prompts, while Turbo is often preferred for cartoon and painting styles. FLUX-15 (FLUX at 15 steps) appears to be favored for general prompts, especially those involving humans.

Ian Goodfellow, Jean Pouget-Abadie, Mehdi Mirza, Bing Xu, David Warde-Farley, Sherjil Ozair, Aaron Courville, and Yoshua Bengio. Generative adversarial nets. *Advances in neural information processing systems*, 27, 2014.

Neha Gupta, Harikrishna Narasimhan, Wittawat Jitkrittum, Ankit Singh Rawat, Aditya Krishna Menon, and Sanjiv Kumar. Language model cascades: Token-level uncertainty and beyond. In *The Twelfth International Conference on Learning Representations*, 2024. URL https://openreview.net/forum?id=KgaBScZ4VI.

Jian Han, Jinlai Liu, Yi Jiang, Bin Yan, Yuqi Zhang, Zehuan Yuan, Bingyue Peng, and Xiaobing Liu. Infinity: Scaling bitwise autoregressive modeling for high-resolution image synthesis, 2024. URL https:

//arxiv.org/abs/2412.04431.

Martin Heusel, Hubert Ramsauer, Thomas Unterthiner, Bernhard Nessler, and Sepp Hochreiter. GANs trained by a two time-scale update rule converge to a local nash equilibrium. In *Advances in Neural Information Processing Systems (NeurIPS)*, volume 30, pp. 6626–6637, 2017.

Jonathan Ho, Ajay Jain, and Pieter Abbeel. Denoising diffusion probabilistic models. *Advances in neural information processing systems*, 33:6840–6851, 2020.

Jonathan Ho, Chitwan Saharia, William Chan, David J Fleet, Mohammad Norouzi, and Tim Salimans. Cascaded diffusion models for high fidelity image generation. *Journal of Machine Learning Research*, 23 (47):1–33, 2022.

Qitian Jason Hu, Jacob Bieker, Xiuyu Li, Nan Jiang, Benjamin Keigwin, Gaurav Ranganath, Kurt Keutzer, and Shriyash Kaustubh Upadhyay. Routerbench: A benchmark for multi-LLM routing system. In *Agentic Markets Workshop at ICML 2024*, 2024. URL https://openreview.net/forum?id=IVXmV8Uxwh.

Dongfu Jiang, Xiang Ren, and Bill Yuchen Lin. Llm-blender: Ensembling large language models with pairwise comparison and generative fusion. In *Proceedings of the 61th Annual Meeting of the Association for Computational Linguistics (ACL 2023)*, 2023.

Wittawat Jitkrittum, Neha Gupta, Aditya Krishna Menon, Harikrishna Narasimhan, Ankit Singh Rawat, and Sanjiv Kumar. When does confidence-based cascade deferral suffice? In *Thirty-seventh Conference on Neural Information Processing Systems*, 2023. URL https://openreview.net/forum?id=4KZhZJSPYU.

Wittawat Jitkrittum, Harikrishna Narasimhan, Ankit Singh Rawat, Jeevesh Juneja, Zifeng Wang, Chen-Yu Lee, Pradeep Shenoy, Rina Panigrahy, Aditya Krishna Menon, and Sanjiv Kumar. Universal model routing for efficient llm inference, 2025. URL https://arxiv.org/abs/2502.08773.

Lynn H Kaack, Priya L Donti, Emma Strubell, George Kamiya, Felix Creutzig, and David Rolnick. Aligning artificial intelligence with climate change mitigation. *Nature Climate Change*, 12(6):518–527, 2022.

Tero Karras, Miika Aittala, Timo Aila, and Samuli Laine. Elucidating the design space of diffusion-based generative models. *Advances in Neural Information Processing Systems*, 35:26565–26577, 2022.

Kate Crawford. Generative ai's environmental costs are soaring — and mostly secret. *Nature World View*, 2024.

Diederik P Kingma and Max Welling. Auto-encoding variational bayes, 2022. URL https://arxiv.org/abs/1312.6114.

Steven Kolawole, Don Dennis, Ameet Talwalkar, and Virginia Smith. Agreement-based cascading for efficient inference, 2024. URL https://arxiv.org/abs/2407.02348.

Lijiang Li, Huixia Li, Xiawu Zheng, Jie Wu, Xuefeng Xiao, Rui Wang, Min Zheng, Xin Pan, Fei Chao, and Rongrong Ji. Autodiffusion: Training-free optimization of time steps and architectures for automated diffusion model acceleration. In *Proceedings of the IEEE/CVF International Conference on Computer Vision*, pp. 7105–7114, 2023a.

Wenhao Li, Xiu Su, Shan You, Tao Huang, Fei Wang, Chen Qian, and Chang Xu. Not all steps are equal: efficient generation with progressive diffusion models. *arXiv preprint arXiv:2312.13307*, 2023b.

Yanyu Li, Huan Wang, Qing Jin, Ju Hu, Pavlo Chemerys, Yun Fu, Yanzhi Wang, Sergey Tulyakov, and Jian Ren. Snapfusion: Text-to-image diffusion model on mobile devices within two seconds. *Advances in Neural Information Processing Systems*, 36, 2024.

Shanchuan Lin, Anran Wang, and Xiao Yang. Sdxl-lightning: Progressive adversarial diffusion distillation. *ArXiv*, abs/2402.13929, 2024. URL https://api.semanticscholar.org/CorpusID:267770548.

Tsung-Yi Lin, Michael Maire, Serge Belongie, James Hays, Pietro Perona, Deva Ramanan, Piotr Dollár, and C Lawrence Zitnick. Microsoft COCO: Common objects in context. In *Computer Vision–ECCV 2014: 13th European Conference, Zurich, Switzerland, September 6-12, 2014, Proceedings, Part V 13*, pp. 740–755. Springer, 2014.

Luping Liu, Yi Ren, Zhijie Lin, and Zhou Zhao. Pseudo numerical methods for diffusion models on manifolds. *arXiv preprint arXiv:2202.09778*, 2022.

Xingchao Liu, Xiwen Zhang, Jianzhu Ma, Jian Peng, et al. Instaflow: One step is enough for high-quality diffusion-based text-to-image generation. In *The Twelfth International Conference on Learning Representations*, 2023.

Cheng Lu, Yuhao Zhou, Fan Bao, Jianfei Chen, Chongxuan Li, and Jun Zhu. Dpm-solver: A fast ode solver for diffusion probabilistic model sampling in around 10 steps. *Advances in Neural Information Processing Systems*, 35:5775–5787, 2022.

Simian Luo, Yiqin Tan, Longbo Huang, Jian Li, and Hang Zhao. Latent consistency models: Synthesizing high-resolution images with few-step inference. *arXiv preprint arXiv:2310.04378*, 2023.

Xinyin Ma, Gongfan Fang, Michael Bi Mi, and Xinchao Wang. Learning-to-cache: Accelerating diffusion transformer via layer caching. *Advances in Neural Information Processing Systems*, 37:133282–133304, 2024a.

Xinyin Ma, Gongfan Fang, and Xinchao Wang. Deepcache: Accelerating diffusion models for free. In *The IEEE/CVF Conference on Computer Vision and Pattern Recognition*, 2024b.

Anqi Mao, Christopher Mohri, Mehryar Mohri, and Yutao Zhong. Two-stage learning to defer with multiple experts. In *Thirty-seventh Conference on Neural Information Processing Systems*, 2023. URL https://openreview.net/forum?id=GIlsH0T4b2.

Chenlin Meng, Robin Rombach, Ruiqi Gao, Diederik Kingma, Stefano Ermon, Jonathan Ho, and Tim Salimans. On distillation of guided diffusion models. In *Proceedings of the IEEE/CVF Conference on Computer Vision and Pattern Recognition*, pp. 14297–14306, 2023.

Hussein Mozannar and David Sontag. Consistent estimators for learning to defer to an expert. In *International conference on machine learning*, pp. 7076–7087. PMLR, 2020.

Hussein Mozannar, Hunter Lang, Dennis Wei, Prasanna Sattigeri, Subhro Das, and David Sontag. Who should predict? exact algorithms for learning to defer to humans. In *International conference on artificial intelligence and statistics*, pp. 10520–10545. PMLR, 2023.

Harikrishna Narasimhan, Wittawat Jitkrittum, Aditya Krishna Menon, Ankit Singh Rawat, and Sanjiv Kumar. Post-hoc estimators for learning to defer to an expert. In Alice H. Oh, Alekh Agarwal, Danielle Belgrave, and Kyunghyun Cho (eds.), *Advances in Neural Information Processing Systems*, 2022. URL https://openreview.net/forum?id=_jg6Sf6tuF7.

Alexander Quinn Nichol and Prafulla Dhariwal. Improved denoising diffusion probabilistic models. In *International conference on machine learning*, pp. 8162–8171. PMLR, 2021.

Isaac Ong, Amjad Almahairi, Vincent Wu, Wei-Lin Chiang, Tianhao Wu, Joseph E. Gonzalez, M Waleed Kadous, and Ion Stoica. RouteLLM: Learning to route LLMs from preference data. In *The Thirteenth International Conference on Learning Representations*, 2025. URL https://openreview.net/forum?id=8sSqNntaMr.

Roni Paiss, Ariel Ephrat, Omer Tov, Shiran Zada, Inbar Mosseri, Michal Irani, and Tali Dekel. Teaching clip to count to ten. *arXiv preprint arXiv:2302.12066*, 2023.

Zizheng Pan, Bohan Zhuang, De-An Huang, Weili Nie, Zhiding Yu, Chaowei Xiao, Jianfei Cai, and Anima Anandkumar. T-stitch: Accelerating sampling in pre-trained diffusion models with trajectory stitching. *arXiv preprint arXiv:2402.14167*, 2024.

Sylvain Paris, Samuel W Hasinoff, and Jan Kautz. Local laplacian filters: Edge-aware image processing with a laplacian pyramid. *ACM Trans. Graph.*, 30(4):68, 2011.

Dustin Podell, Zion English, Kyle Lacey, Andreas Blattmann, Tim Dockhorn, Jonas Müller, Joe Penna, and Robin Rombach. Sdxl: Improving latent diffusion models for high-resolution image synthesis, 2023. URL https://arxiv.org/abs/2307.01952.

Konpat Preechakul, Nattanat Chatthee, Suttisak Wizadwongsa, and Supasorn Suwajanakorn. Diffusion autoencoders: Toward a meaningful and decodable representation. In *Proceedings of the IEEE/CVF Conference on Computer Vision and Pattern Recognition*, pp. 10619–10629, 2022.

Alec Radford, Jong Wook Kim, Chris Hallacy, Aditya Ramesh, Gabriel Goh, Sandhini Agarwal, Girish Sastry, Amanda Askell, Pamela Mishkin, Jack Clark, et al. Learning transferable visual models from natural language supervision. In *International conference on machine learning*, pp. 8748–8763. PMLR, 2021.

Aditya Ramesh, Mikhail Pavlov, Gabriel Goh, Scott Gray, Chelsea Voss, Alec Radford, Mark Chen, and Ilya Sutskever. Zero-shot text-to-image generation. In *International conference on machine learning*, pp. 8821–8831. Pmlr, 2021.

Robin Rombach, Andreas Blattmann, Dominik Lorenz, Patrick Esser, and Björn Ommer. High-resolution image synthesis with latent diffusion models. In *Proceedings of the IEEE/CVF conference on computer vision and pattern recognition*, pp. 10684–10695, 2022.

Chitwan Saharia, William Chan, Saurabh Saxena, Lala Li, Jay Whang, Emily Denton, Seyed Kamyar Seyed Ghasemipour, Burcu Karagol Ayan, S. Sara Mahdavi, Rapha Gontijo Lopes, Tim Salimans, Jonathan Ho, David J Fleet, and Mohammad Norouzi. Photorealistic text-to-image diffusion models with deep language understanding, 2022. URL https://arxiv.org/abs/2205.11487.

Tim Salimans and Jonathan Ho. Progressive distillation for fast sampling of diffusion models. *arXiv preprint arXiv:2202.00512*, 2022.

Sara Sangalli, Ertunc Erdil, and Ender Konukoglu. Expert load matters: operating networks at high accuracy and low manual effort. *Advances in Neural Information Processing Systems*, 36:16283–16301, 2023.

Sarah Wells. Generative ai's energy problem today is foundational. *IEEE Spectrum*, 2023.

Axel Sauer, Frederic Boesel, Tim Dockhorn, Andreas Blattmann, Patrick Esser, and Robin Rombach. Fast high-resolution image synthesis with latent adversarial diffusion distillation. *arXiv preprint arXiv:2403.12015*, 2024.

Christoph Schuhmann, Romain Beaumont, Richard Vencu, Cade Gordon, Ross Wightman, Mehdi Cherti, Theo Coombes, Aarush Katta, Clayton Mullis, Mitchell Wortsman, Patrick Schramowski, Srivatsa Kundurthy, Katherine Crowson, Ludwig Schmidt, Robert Kaczmarczyk, and Jenia Jitsev. Laion-5b: an open large-scale dataset for training next generation image-text models. In *Proceedings of the 36th International Conference on Neural Information Processing Systems*, NIPS '22, Red Hook, NY, USA, 2022. Curran Associates Inc. ISBN 9781713871088.

Jiaming Song, Chenlin Meng, and Stefano Ermon. Denoising diffusion implicit models. *arXiv preprint arXiv:2010.02502*, 2020.

Yang Song, Prafulla Dhariwal, Mark Chen, and Ilya Sutskever. Consistency models. *arXiv preprint arXiv:2303.01469*, 2023.

Dharmesh Tailor, Aditya Patra, Rajeev Verma, Putra Manggala, and Eric Nalisnick. Learning to defer to a population: A meta-learning approach, 2024. URL https://arxiv.org/abs/2403.02683.

Shengkun Tang, Yaqing Wang, Caiwen Ding, Yi Liang, Yao Li, and Dongkuan Xu. Adadiff: Accelerating diffusion models through step-wise adaptive computation. *arXiv preprint arXiv:2309.17074*, 2023.

Ashish Vaswani, Noam Shazeer, Niki Parmar, Jakob Uszkoreit, Llion Jones, Aidan N Gomez, Łukasz Kaiser, and Illia Polosukhin. Attention is all you need. *Advances in neural information processing systems*, 30, 2017.

Xiaofang Wang, Dan Kondratyuk, Eric Christiansen, Kris M. Kitani, Yair Movshovitz-Attias, and Elad Eban. Wisdom of committees: An overlooked approach to faster and more accurate models. In *International Conference on Learning Representations*, 2022a. URL https://openreview.net/forum?id=MvO2t0vbs4-.

Zijie J. Wang, Evan Montoya, David Munechika, Haoyang Yang, Benjamin Hoover, and Duen Horng Chau. DiffusionDB: A large-scale prompt gallery dataset for text-to-image generative models. *arXiv:2210.14896 [cs]*, 2022b. URL https://arxiv.org/abs/2210.14896.

Xiaoshi Wu, Yiming Hao, Keqiang Sun, Yixiong Chen, Feng Zhu, Rui Zhao, and Hongsheng Li. Human preference score v2: A solid benchmark for evaluating human preferences of text-to-image synthesis. *arXiv preprint arXiv:2306.09341*, 2023.

Jiazheng Xu, Xiao Liu, Yuchen Wu, Yuxuan Tong, Qinkai Li, Ming Ding, Jie Tang, and Yuxiao Dong. Imagereward: learning and evaluating human preferences for text-to-image generation. In *Proceedings of the 37th International Conference on Neural Information Processing Systems*, pp. 15903–15935, 2023.

Shuai Yang, Yukang Chen, Luozhou Wang, Shu Liu, and Yingcong Chen. Denoising diffusion step-aware models. *arXiv preprint arXiv:2310.03337*, 2023.

Han Zhang, Tao Xu, Hongsheng Li, Shaoting Zhang, Xiaogang Wang, Xiaolei Huang, and Dimitris N Metaxas. Stackgan: Text to photo-realistic image synthesis with stacked generative adversarial networks. In *Proceedings of the IEEE international conference on computer vision*, pp. 5907–5915, 2017.

Hui Zhang, Zuxuan Wu, Zhen Xing, Jie Shao, and Yu-Gang Jiang. Adadiff: Adaptive step selection for fast diffusion. *ArXiv*, abs/2311.14768, 2023. URL https://api.semanticscholar.org/CorpusID:265456553.

Richard Zhuang, Tianhao Wu, Zhaojin Wen, Andrew Li, Jiantao Jiao, and Kannan Ramchandran. EmbedLLM: Learning compact representations of large language models. In *The Thirteenth International Conference on Learning Representations*, 2025. URL https://openreview.net/forum?id=Fs9EabmQrJ.

# Cost-Aware Routing for Efficient Text-To-Image Generation
## Appendix

## A Human Preference Score

To quantitatively evaluate how our routing decisions from each metric align with human perception, we adopt the Human Preference Score v2 (HPSv2) benchmark (Wu et al., 2023). We trained four separate routers, one for each metric (CLIPScore, Sharpness, Aesthetic Score, and ImageReward), and then report the average HPSv2 quality score achieved by our routing approach when operating at the cost (TFLOPs) of each model in the pool. For this analysis, we also expand our candidate pool to include the recent, state-of-the-art FLUX model (FLUX.1-dev with 30 denoising steps)[3]. The results are shown below.

| HPSv2 score ↑ | | | | | | | | | | |
|---|---|---|---|---|---|---|---|---|---|---|
| | Infinity | Turbo | Lighting | SDXL-10 | DeepCache | SDXL-20 | DDIM | SDXL-50 | SDXL100 | FLUX |
| Fixed Baseline | 0.293 | 0.288 | 0.304 | 0.251 | 0.277 | 0.275 | 0.284 | 0.284 | 0.286 | 0.298 |
| Ours (CLIPScore) | 0.293 | 0.288 | 0.287 | 0.282 | 0.277 | 0.278 | 0.284 | 0.284 | 0.286 | 0.298 |
| Ours (Sharpness) | 0.293 | 0.293 | 0.293 | 0.293 | 0.293 | 0.293 | 0.295 | 0.295 | 0.296 | 0.298 |
| Ours (ImageReward) | 0.293 | 0.293 | 0.293 | 0.293 | 0.293 | 0.293 | 0.295 | 0.295 | 0.296 | 0.298 |
| Ours (Aesthetic) | 0.293 | 0.293 | 0.304 | 0.304 | 0.303 | 0.303 | 0.302 | 0.302 | 0.300 | 0.298 |
| **TFLOPs** | 1.50 | 1.54 | 23.92 | 119.70 | 210.00 | 239.40 | 598.50 | 598.50 | 1197.00 | 1785.60 |

Notably, the results indicate that routing with the Aesthetic Score produces outcomes that correlate most closely with human preferences as evaluated by HPSv2. This is likely because the Aesthetic Score model itself is trained on human ratings of aesthetic value. Note that when the cost constraint is set to match the cheapest (Infinity) or most expensive (FLUX) model, the router must select that model for all prompts, resulting in identical performance.

## B Additional Qualitative Analysis

In Figure 5, we analyze scenarios showing both successes and failures of our adaptive routing method (*Proposed Transformer (SDXL+)* on CLIPScore metric). Specifically, we focus on cases where our method uses the same overall computational cost as the baseline (SDXL with a fixed 22 denoising steps). Within these scenarios, we consider cases where our method allocates more than 22 denoising steps, indicating that the prompts are particularly complex and require additional refinement.

For the prompt *A young kid stands before a birthday cake decorated with Captain America*, our method correctly recommends more denoising steps, as fewer would not generate accurate images. In contrast, the prompt *There are two traffic signals on a metal pole, each with three light signals on them* includes an exact number of objects, a concept which both diffusion models and CLIP often struggle with (Binyamin et al., 2024; Paiss et al., 2023). Our approach accounts for this difficulty by recommending more steps than average. However, in this case, more denoising steps actually degrade image quality which is uncommon and ends up hurting the router performance.

We also perform a user study to compare the subset of these routing decisions with the fixed cost baseline (see Appendix Section I). All participants rate Figure 5b (ours) as the better image, while 14 of 19 participants select Figure 5c (baseline) as the better image.

## C Cross Dataset Generalization

To evaluate cross-dataset generalization, we conducted an experiment where our quality estimator (KNN) was trained on DiffusionDB prompts and evaluated on the prompts from COCO captioning dataset. This is a

---

[3]https://huggingface.co/black-forest-labs/FLUX.1-dev

**Success case**. A young kid stands before a birthday cake decorated with Captain America

**Failure case**. There are two traffic signals on a metal pole, each with three light signals on them.

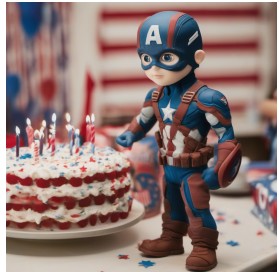
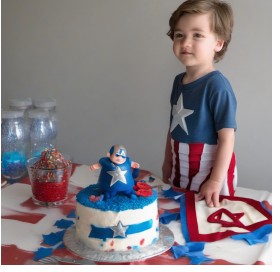
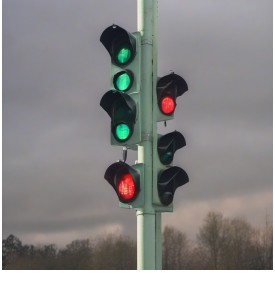
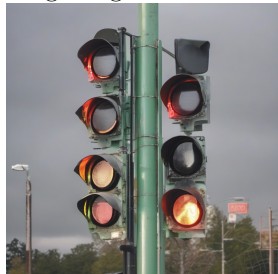

| (a) 22 steps (fixed) | (b) 27 steps (routed) | (c) 22 steps (fixed) | (d) 27 steps (routed) |

Figure 5: Success and failure cases of the baseline SDXL with static 22 denoising steps, and our approach *Proposed Transformer (SDXL+) in Figure 3a* operating at the same average cost as the baseline. **(a)**, **(b)**: Our approach is able to recognize the need for a larger number of denoising steps to generate an image that matches the prompt. **(c)**, **(d)**: Prompts that specify an exact number of objects are difficult for diffusion models in general. The number of objects may fluctuate during the denoising process, making it difficult to predict the right number of steps.

challenging test due to the significant stylistic differences between the datasets. In Section C, we report the average quality (Sharpness score) achieved by our routing approach when operating at the cost (TFLOPs) of each model in the pool. Our framework still achieves a good quality-cost trade-off even without being trained on this new prompt distribution. This suggests the estimator learns a robust understanding of prompt complexity beyond simple keyword correlation.

| **Sharpness ↑** | | | | | |
|---|---|---|---|---|---|
| | Turbo | Lighting | DeepCache | DDIM | SDXL-100 |
| Ours | 0.1152 | 0.1150 | 0.1134 | 0.1100 | 0.1048 |
| Fixed | 0.1152 | 0.1069 | 0.0979 | 0.1049 | 0.1048 |

## D   Model Architecture

In this section, we provide more details on the architecture of the models we used for the quality estimator $\hat{\gamma}^m$ in Eq. (3).

**$K$-NN**   The $K$-NN approach provides a non-parametric way to estimate quality by retrieving and averaging the quality scores of $K$ nearest training prompts, in an appropriate embedding space. This method is simple, and can generalize well with sufficient data. As no iterative training is required for $K$-NN (besides producing a search index), it can be a suitable model when the underlying pool $\mathcal{H}$ of base models changes frequently. We use the CLIP text encoder Radford et al. (2021) to produce prompt embeddings The text encoder is used directly without fine-tuning.

**Transformer**   Our Transformer-based estimator is built on top of the text embeddings produced by the CLIP text encoder. Specifically, for each prompt, CLIP considers the first 77 tokens and produces 77 per-token embeddings, each of 768 dimensions. To construct our quality estimation model, we first add two self-attention layers with position embeddings, resulting in an output sequence of 77 per-token embeddings, each of 512 dimensions. We then add a small output head with a 2-layer linear MLP with a Sigmoid activation function on top of each of these token embeddings. Averaging across the tokens produces $M$ scores $\hat{\gamma}^{(1)}(\mathbf{x}), \ldots, \hat{\gamma}(\mathbf{x})^{(M)}$ (see Eq. (3)), each estimating the expected quality of the $m$-th model on prompt $\mathbf{x}$.

We train a separate model for each of the quality metrics considered. In each case, the quality scores are linearly normalized across all training examples to be in [0, 1]. These normalized metrics are treated

as ground-truth probabilities, and the model is trained by minimizing the cross-entropy losses. Only the attention layers and the MLP are trained with the frozen CLIP text encoder.

Note that all the base models except Infinity consume the CLIP text embedding as input. Thus, the cost of invoking our router is only from the extra layers added in the case of Transformer, or neighbor lookup in the case of $K$-NN. The overhead in terms of FLOPs is negligible compared to invoking SDXL. Since the Infinity baseline uses Flan-T5 instead of CLIP, this incurs an additional $\sim 13.087$ GFLOPs if Infinity is selected. To put it in perspective, calling SDXL for one prompt with 17 denoising steps would incur roughly 200 TFLOPs.

## E    Computational Resources

To generate our training set (quality score per prompt), we used 50 A100/H100 80G GPU with approximately $\sim$4 days per model to generate 391,364 images for the filtered DiffusionDB prompt set (97,841 prompts) and less than 1 day per model for the filtered COCO prompt set (18,384 prompts). For each quality metric, we trained our Transformer with one A100 40G GPU in $\sim$2 hours for the COCO prompt set and $\sim$7 hours for the DiffusionDB prompt set. The trained transformer (33.88M parameters, 15.61 GFLOPs) represents just 1.7% of the size and 1.04% of the computational cost relative to the smallest candidate, Infinity (2B parameters, 1.5 TFLOPs). For our kNN-based router, the overhead is even more marginal, as it only requires a single-pass search over a small set of $10^3$–$10^4$ reference samples. The trained Transformer model only takes $\sim$0.05 seconds to predict the scores for a single prompt in one A100 40G GPU. We trained KNN on the CPU in less than 1 minute with negligible inference time($<$0.01 second per prompt).

## F    Quality-Neutral Costs

We provide quantitative metrics to complement the deferral curves shown in Figure 3a (COCO dataset, evaluated with CLIPScore). We consider the quality-neutral cost (QNC) Ong et al. (2025); Jitkrittum et al. (2025) defined as the fraction of cost required to reach the performance of a reference model. The lower the QNC, the better because this means that our method can achieve the same performance as a reference model using a lower cost. The QNCs of the two proposed methods are shown in the following table, where the reference is set to each of the individual model in the pool (described in Section 5.1).

| Method   \   QNC (%) | INFINITY | DDIM | DEEPCACHE | LIGHTING | TURBO | SDXL100 |
|---|---|---|---|---|---|---|
| *Transformer (SDXL+)* | 100 | 2.5 | 11.1 | 6.4 | 99.7 | 2.1 |
| *K-NN (SDXL+)* | 100 | 2.5 | 12.3 | 6.4 | 100.2 | 2.4 |

For example, a QNC of 100% to INFINITY indicates that our approaches would need the full cost of INFINITY to reach its average performance. In other words, visually, the deferral curves of the two proposed methods would pass through the quality-cost operating point of INFINITY. As another example, the QNC of the *Proposed Transformer (SDXL+)* relative to the baseline DEEPCACHE is 11.1%, indicating that *Proposed Transformer (SDXL+)* only needs 11.1% of the cost of DEEPCACHE to have the same performance.

Overall, our proposed approaches are able to match the quality of all the baselines with either a significantly lower cost, or almost the same cost.

# G    Model Selection Rates

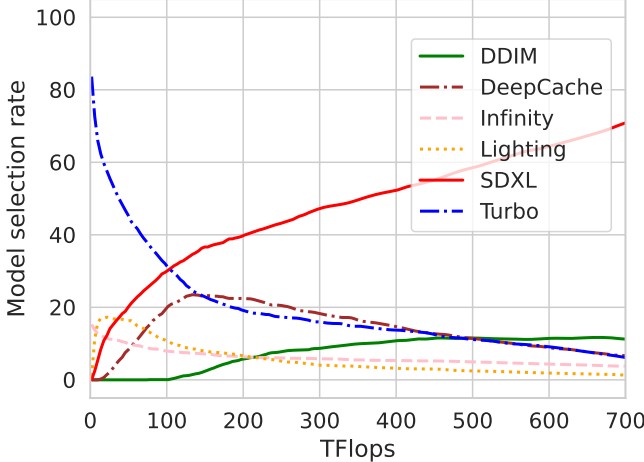

Figure 6: The rate at which each choice in the candidate routing pool is selected by *Proposed Transformer (SDXL+)* in Figure 3a. Our approach is able to adaptively mix and match different model choices throughout the cost range.

Figure 6 shows the rate at which each choice in $\mathcal{H}$ is selected by *Proposed Transformer (SDXL+)* in Figure 3a. All the 12 candidate diffusion steps offered by the base SDXL model are collapsed into one curve for clarity. We observe that, when the cost budget is large, the router increasingly allocates resources to the full SDXL model. On the other hand, in the lower cost range, TURBO is the prominent choice, as it provides a good balance between cost and quality. This analysis shows that our router is able to adaptively mix and match different choices throughout the cost range to achieve a good quality-cost trade-off.

# H    $K$-NN Parameter Selection

Figure 7 shows the deferral curve when using $K$-NN as a quality estimator at various values of $K$. As shown in the figure, the routing performance (on a validation set of 828 prompts drawn from the COCO dataset) is similar across a wide range of $K$ values. We set $K = 100$ for our final model.

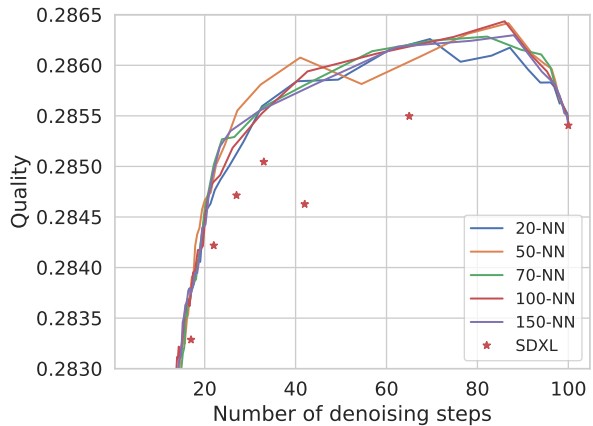

Figure 7: Performance of the proposed $K$-NN-based routing model on a validation subset drawn from COCO.

# I    User Study

Here we provide details on the user study to evaluate our routing decision. For this qualitative analysis, we consider the same trained router shown in Figure 3a as *Proposed Transformer (SDXL+)*. Our reference baseline is the SDXL model with the number of denoising steps set to 22; this setting results in a per-prompt cost of 263.3 TFLOPs. For a fair comparison, we accordingly consider the operating point of our method that has the same average cost as this baseline by adjusting $\lambda$ in Eq. (3). Of all the prompts in the test set used in Figure 3a, we consider a random subset of 100 prompts where our method does not select SDXL with 22 denoising steps as the routed decision; this filtering is done to facilitate a contrast between the two approaches. We proceeded to recruit 19 participants through the *Prolific* platform for a human preference analysis (https://www.prolific.com). We run a two-alternative forced-choice (2AFC) study to measure participants' preference for images produced by both approaches. During each trial, each participant is presented with a text prompt, and two test images produced by the two approaches. The participant is instructed to select the image that better matches the prompt.

**Protocol** For each trial, a prompt is shown first to each participant. Two randomized-order images are then presented on participants' screens: 1) image from our *Proposed Transformer (SDXL+)* (in Figure 3a), and 2) image produced by the baseline SDXL at fixed 22 denoising steps. Note that the produced image may be from a non-SDXL model (e.g., Turbo) since our approach may route to other baseline models described in Section 5.1.

Participants were instructed to select the image that matches the input text prompt better. Each participant will be assigned 100 trials (100 prompts) in total, with prompts randomly sampled from the COCO dataset described in Section 5.2. Our crowdsourcing user study protocol is visualized in Figure 8 as a sequence of web pages that will be shown to participants with example stimuli.

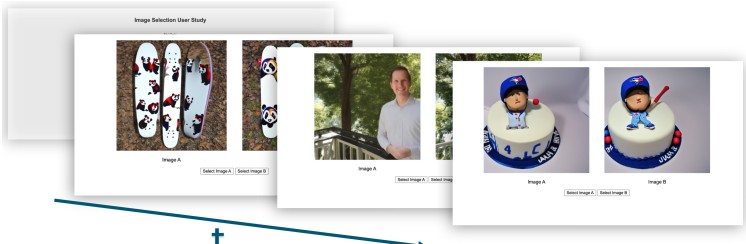

Figure 8: *User study protocol Ours vs. SDXL with static 22 denoising steps.* In each user study trial, the participant will see two images: Image A and Image B. The task is to select the image that match better with the text prompt provided. The participant needs to click on the button below or press the keyboard to choose A/B.

**Additional Results**

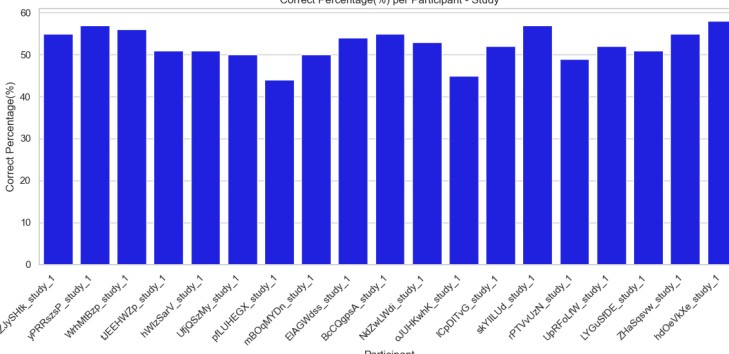

Figure 9: The rate at which each participant prefers our suggested image over the image produced by the baseline SDXL with 22 denoising steps.

Figure 9 shows the rate, in percentage, at which each participant selects our *Proposed Transformer (SDXL+)*. Here, we define **percentage selection** as the proportion of trials in which ours is selected. We note that for Stable Diffusion XL, many images generated from 22 denoising steps already show reasonable results with hard-to-notice artifacts, which can limit the perceivable differences. In the main paper, we highlight that the majority of participants only agreed on those most notable cases. On average, the participants prefer ours 52.37% of the time.

## J    Comparison with DeepCache on Quality-Cost Trade-off

Figure 10 compares our adaptive routing method (*Proposed Transformer (SDXL+)*) on CLIPScore with the DeepCache approach Ma et al. (2024b) (on SDXL model at 50 denoising steps). DeepCache caches intermediate activations at predefined intervals (cache intervals) to balance image quality and computational cost. Varying the cache interval enables different quality–cost trade-offs, which can then be compared with our method on a deferral curve. As shown in Figure 10, our method, which adaptively utilizes multiple models, consistently surpasses fixed-interval caching methods across all computational costs. Note that our adaptive routing strategy can also incorporate any DeepCache configurations to even further enhance performance.

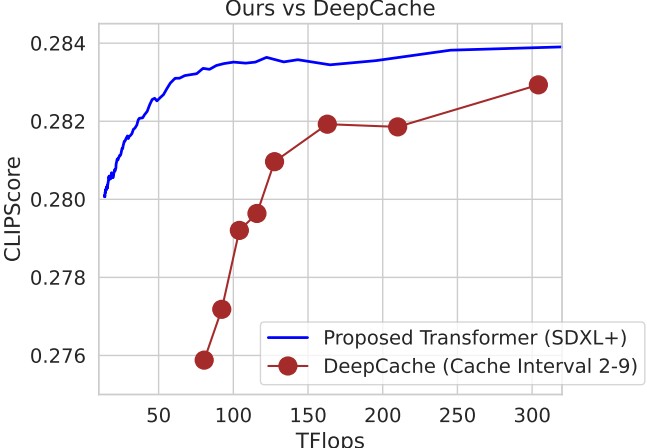

Figure 10: CLIPScore-TFlops trade off comparison with Deepcache at various cache interval on the test set of COCO dataset

## K    Statistical Significance of COCO Results

Recall that Figure 3 shows the deferral curves of our methods on COCO dataset. To give a more precise view on the performance improvement, in Table 5, we report average quality scores attained by our *Proposed K-NN (SDXL+)* at the same costs as the individual models in the pool. We observe that at the operating cost of each individual model in the pool, our approach is able to deliver a higher average quality score (as measured by CLIPScore and Sharpness). In most cases, the gains are statistically significantly better as warranted by the Welch's t-test. Note that at the two extreme ends of operating costs (i.e., calling the cheapest and most expensive models, respectively), any routing approach necessarily reduces to trivial routing: sending all prompts to one model. It follows that, at an extreme operating point (either at the lowest possible or highest possible cost), the average quality achieved must be exactly the same as that of the individual model at that end point.

Table 5: Quality–cost breakdown for *Proposed K-NN (SDXL+)* presented in Table 5 (on COCO dataset). An entry in bold text indicates that, with the same cost, our approach is statistically significantly better than the corresponding individual model (Welch's t-test at significance level $\alpha = 0.05$).

| **CLIPScore** Radford et al. (2021) ↑ | | | | | | | | |
|---|---|---|---|---|---|---|---|---|
| Ours | 0.2672 | 0.2801 | **0.2817** | **0.2830** | **0.2832** | **0.2832** | **0.2832** | 0.2830 | 0.2822 |
| Fixed | 0.2672 | 0.2798 | 0.2749 | 0.2773 | 0.2820 | 0.2805 | 0.2810 | 0.2816 | 0.2822 |
| **Sharpness** ↑ | | | | | | | | |
| Ours | 0.1061 | 0.1161 | **0.1166** | **0.1163** | **0.1160** | **0.1158** | **0.1132** | **0.1121** | 0.1048 |
| Fixed | 0.1061 | 0.1152 | 0.1070 | 0.0997 | 0.0979 | 0.0858 | 0.0995 | 0.1049 | 0.1048 |
| Model | Infinity | Turbo | Lighting | SDXL-9 | DeepCache | SDXL-22 | SDXL-42 | DDIM | SDXL100 |
| Cost | 1.50 | 1.54 | 23.92 | 107.73 | 210.00 | 263.34 | 502.74 | 598.50 | 1197.00 |

## L  Additional Deferral Curve

Here we show the complete deferral curves for the four quality metrics–CLIPScore, Sharpness, ImageReward, and Aesthetic Score, on the test set of DiffusionDB dataset in Figure 11. These curves complement the fixed-cost comparison in Table 2 by showing the changes in quality score across the entire cost spectrum. Here we can clearly see how our adaptive routing consistently achieves a higher quality score than the fixed-model baselines at every computational budget.

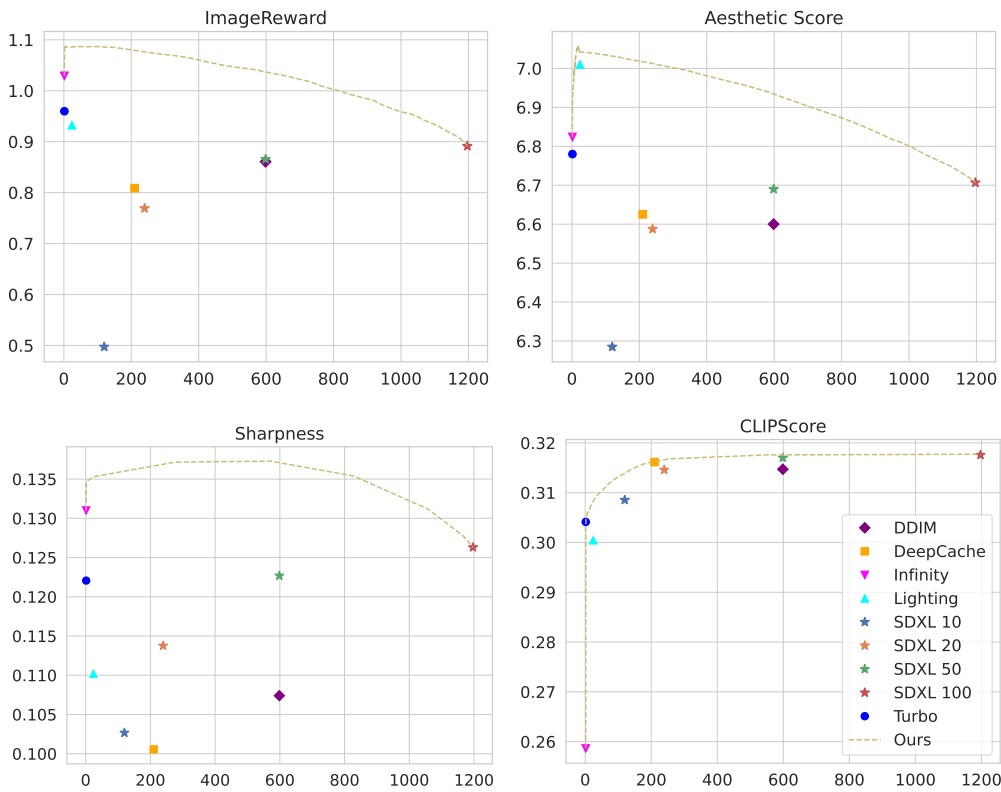

Figure 11: Deferral Curves of our *Proposed Transformer (SDXL+)* on DiffusionDB dataset. Our approach exceeds the quality of fix-step text-to-image models in all quality metrics (ImageReward, Aesthetic, Sharpness, and CLIPScore).

## M   Estimation Errors and Quality-Cost Trade-offs

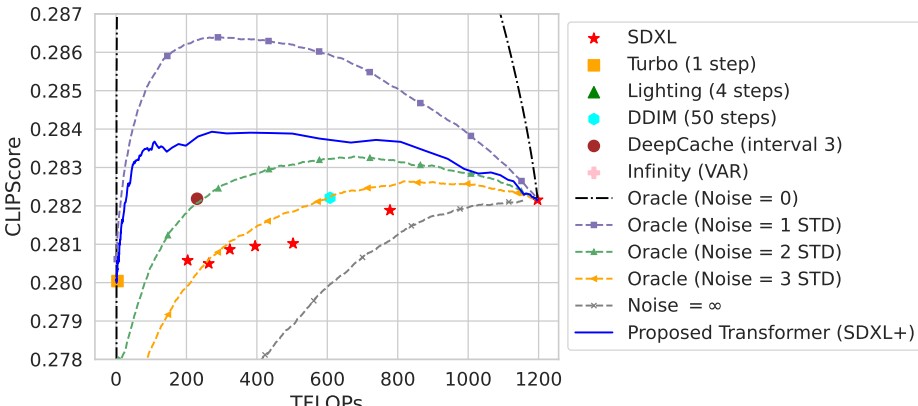

Figure 12: The deferral curves as presented in Figure 3a, supplemented with baselines derived from the oracle routing rule with noise added to simulate routing errors. See Section M for details.

Table 6: Average quality scores of the routing approaches presented in Figure 12.

**CLIPScore ↑**

|  | Infinity | Turbo | Lighting | SDXL-9 | DeepCache | SDXL-22 | DDIM | SDXL-65 |
|---|---|---|---|---|---|---|---|---|
| Ours | .2672 | .2800 | .2820 | .2836 | .2837 | .2840 | .2838 | .2837 |
| Fixed | .2672 | .2798 | .2749 | .2773 | .2820 | .2805 | .2816 | .2819 |
| Oracle | .2672 | .2855 | .2928 | .2973 | .2984 | .2986 | .2969 | .2944 |
| Oracle (Noise = 1 STD) | .2672 | .2806 | .2829 | .2854 | .2862 | .2864 | .2860 | .2851 |
| Oracle (Noise = 2 STD) | .2672 | .2780 | .2782 | .2806 | .2819 | .2823 | .2832 | .2832 |
| Oracle (Noise = 3 STD) | .2672 | .2768 | .2763 | .2784 | .2800 | .2806 | .2822 | .2825 |
| Random (Noise = ∞) | .2672 | .2727 | .2711 | .2733 | .2756 | .2763 | .2799 | .2811 |
| TFLOPs | 1.5 | 1.54 | 23.92 | 107.73 | 210 | 263.34 | 598.5 | 778.05 |

In this section, we discuss the relationship between estimation errors of a quality metric, and the resulted quality-cost trade-off. Specifically, the goal is to quantify the degradation of the routing performance from the oracle routing rule, as estimation errors increase. To demonstrate this, we consider the same experimental setup as used in Figure 3a i.e., with COCO as the dataset, and with CLIPScore as the image quality metric.

**Plug-in estimate of the oracle rule**   We start with a plug-in empirical estimator $\hat{r}^*$ of the optimal (oracle) routing rule in Theorem 1. Recall that the optimal routing rule in Theorem 1 is given by

$$r^*(\mathbf{x}) = \arg\max_{m \in [M]} \mathbb{E}\left[ q(\mathbf{x}, h^{(m)}(\mathbf{x})) \mid \mathbf{x} \right] - \lambda \cdot c^{(m)}.$$

Recall from Section 3.2 that $\hat{y}_{i,m}$ denotes the empirical estimate of $\mathbb{E}\left[ q(\mathbf{x}_i, h^{(m)}(\mathbf{x})) \mid \mathbf{x}_i \right]$. Define $\hat{\mathbf{y}}_i \doteq (\hat{y}_{i,1}, \dots, \hat{y}_{i,M})$. For a labeled example $(\mathbf{x}_i, \hat{y}_i)$, the optimal routing rule can thus be estimated as

$$\hat{r}^*(\mathbf{x}_i) = \arg\max_{m \in [M]} \hat{y}_{i,m} - \lambda \cdot c^{(m)}. \tag{4}$$

This data-based oracle rule is directly applicable to test examples in the test set without require any estimation. The resulting deferral curve is denoted by "Oracle (Noise = 0)" in Figure 12. Evidently, this routing rule exhibits an excellent quality-cost trade-off compared to other routing approaches. Indeed, it makes use of the ground-truth quality label $\mathbf{y}$ to make a routing decision. By construction, no other routing methods can give

a better trade-off curve than this deferral curve (on this specific dataset). We emphasize that in practice it is extremely challenging to realize a quality-cost operating point that is close to this oracle routing rule. See, for instance, Figure 3 and Figure 4 in Hu et al. (2024) for the performance gap to the oracle in the context of LLM routing (i.e., not routing to text-to-image models, as considered in our work). Nevertheless, the oracle curve serves as an upper bound on the trade-off achievable by any routing methods.

**Noisy oracle** We now consider adding noise to the routing rule in (4) to examine how noise affects the quality-cost trade-off:

$$\hat{r}_\beta^*(\mathbf{x}_i) = \arg \max_{m \in [M]} \hat{y}_{i,m} + \beta \cdot \mathrm{STD}_m \cdot g_{i,m} - \lambda \cdot c^{(m)}, \tag{5}$$

where

- $\beta \geq 0$ controls the strength of Gaussian noise to add;

- $\mathrm{STD}_m \doteq \sqrt{\frac{1}{N_{\mathrm{te}}} \sum_{n=1}^{N_{\mathrm{te}}} (y_{n,m} - \bar{y}_m)^2}$, and $\bar{y}_m \doteq \frac{1}{N_{\mathrm{te}}} \sum_{n=1}^{N_{\mathrm{te}}} y_{n,m}$;

- $N_{\mathrm{te}}$ denotes the number of test examples; and

- $g_{i,m} \overset{\mathrm{i.i.d.}}{\sim} \mathcal{N}(0,1)$ is an independent realization from the standard normal distribution.

Clearly, if $\beta = 0$, we recover the empirical oracle rule in (4). The parameter $\beta$ represents the amount of Gaussian noise added to the quality scores, in the unit of the standard deviation of the quality scores of the respective model.

In Figure 12 and Table 6, we present the performance of the noisy routing rule with $\beta \in \{1, 2, 3, \infty\}$. When $\beta = \infty$, we simply perform random routing, which gives a poor trade-off. It can be seen that our proposed the deferral curve of our proposed router lies between the curves of the oracle with $\beta = 1$ and the oracle with $\beta = 2$. Roughly, this means that our router has a similar performance to an oracle router where the per-model ground-truth scores are corrupted with independent Gaussian noise at a factor $\beta \in (1, 2)$ of their standard deviations.

