# OpenReview forum: "Cost-Aware Routing for Efficient Text-To-Image Generation"
_TMLR — Accepted by TMLR_

### Review · Reviewer_3zX2 · 2025-12-08

**Summary Of Contributions:**

To address the limitations of existing methods in effectively balancing quality and efficiency under fixed inference budgets, this paper proposes CATImage, a cost-aware routing framework for text-to-image generation tasks. First, the authors identify that input prompts exhibit significant variance in their intrinsic generation difficulty, rendering uniform computational allocation suboptimal for balancing overall quality and cost. Consequently, the paper formalizes the problem as an optimization task subject to a computational budget constraint and derives a theoretically optimal Bayesian routing strategy based on this formulation. The framework employs a trained lightweight quality estimator to predict the expected quality of various candidate generators—ranging from homogeneous adjustments of denoising steps to heterogeneous switches between distinct model architectures. During inference, it dynamically selects the most appropriate generation path for each input prompt according to the derived Bayesian optimal rule. Systematic experiments on the COCO and DiffusionDB datasets demonstrate that CATImage establishes a performance curve in the quality-cost plane that outperforms all single static models, significantly reducing average computational costs while maintaining or even enhancing generation quality.

**Additional Comments:**

1. Quantify Router Overhead

Please report the parameter size of the quality estimator and its actual inference latency on the hardware used in the experiments. Additionally, provide the ratio of this latency relative to the inference time of the fastest model in the candidate pool (e.g., SDXL-Turbo or Lightning). If this proportion is significant (e.g., >10%), please revise the relevant claims regarding "negligible overhead" or discuss the implications for real-time applications.

2. Discuss Maintenance Cost for Extending the Candidate Pool

I suggest explicitly clarifying in the Limitations or Discussion section that introducing a new model to the candidate pool requires re-generating quality labels across the entire training set and retraining the router. Please acknowledge that this offline overhead may limit the framework's scalability in scenarios where models are rapidly updated or iterated.

3. Provide Error Rate and Impact Analysis of Misrouting

To provide a more comprehensive understanding of the router's robustness, I suggest reporting the misrouting rate on the test set. Furthermore, please quantify the average degradation in quality caused by these misclassifications compared to the optimal routing choice.

**Audience:**

Yes

**Audience Explanation:**

Insufficient Analysis of Router Overhead: While the overall methodology is clear, the paper lacks a detailed quantitative analysis of the router's own computational overhead. Key metrics such as the actual inference latency and parameter size of the quality estimator are missing, which are crucial for evaluating the net efficiency gains.

High Maintenance Cost for Updates: Although the paper emphasizes the framework's flexibility, the discussion regarding maintenance costs is limited. Introducing a new candidate model requires re-generating images and computing quality scores across the entire training dataset to retrain the router.

Lack of Error Analysis and Confidence Estimation: Each prompt can only be routed to a single model, and an incorrect selection will directly degrade the generation quality. The paper neither analyzes the router's error rate nor its impact on final image quality, nor does it provide uncertainty or confidence estimates.

**Broader Impact Concerns:**

This work is highly relevant to researchers focusing on Efficient ML, generative model acceleration, and deployment optimization. As diffusion models become increasingly ubiquitous in real-world applications, reducing inference costs while maintaining generation quality has emerged as a pervasive technical challenge. The proposed routing framework, grounded in the "Learning-to-Defer" paradigm, offers a novel perspective on enhancing inference efficiency without modifying the underlying model architectures. Consequently, it holds potential value for both researchers and practitioners exploring heterogeneous generative model collaboration and strategies to reduce latency and energy consumption. Furthermore, the discussion regarding the relationship between prompt difficulty and computational requirements may provide further insights into understanding the performance boundaries of diffusion models.

**Claims And Evidence:**

Yes

**Claims Explanation:**

The work is well-motivated and features a methodological design with strong generality, effectively extending the concepts of "Learning-to-Defer" and "Model Routing" to the text-to-image generation domain for the first time. The proposed scheme is model-agnostic, requiring no modifications to the base generative models. By simply training a quality estimator to predict performance and applying the derived cost-aware Bayesian optimal criterion, it dynamically routes input prompts—based on their intrinsic difficulty—to the most cost-effective generation model or number of denoising steps. Consequently, the framework demonstrates strong engineering practicality and scalability, allowing for the seamless integration of more efficient models developed in the future. In terms of experimental design, the use of deferral curves intuitively demonstrates the superiority of dynamic routing over static strategies, rendering the conclusions credible.

**Requested Changes:**

The submission supports its claims through both theoretical grounding and empirical validation. Theoretically, the authors derive an optimal routing rule based on Bayesian decision theory, providing a sound mathematical foundation for the proposed decision-making strategy. Empirically, the authors conduct a systematic evaluation on two representative datasets, COCO and DiffusionDB, benchmarking against multiple strong baselines, including SDXL-Turbo, Lightning, and DeepCache. By presenting detailed "deferral curves," the results demonstrate that CATImage achieves a superior quality-cost trade-off compared to individual generative models across varying computational budgets. Furthermore, the inclusion of ablation studies on homogeneous versus heterogeneous routing schemes, along with an analysis of specific failure cases (e.g., counting-based prompts), further bolsters the empirical findings. Overall, the experimental setup is comprehensive and the comparisons are sufficient, providing robust support for the paper's main methodological claims.

---

### Review · Reviewer_R7ob · 2025-12-22

**Summary Of Contributions:**

This paper presents a framework to adaptively selects the most efficient text to image generation models for a given prompt based on its complexity through cost-aware routing. The major contribution is applying the optimization formulation from Universal Model Routing to text-to-image generation with its specific image quality estimator. It uses the optimal routing rule inferenced from the proposition 1 to route the high-complexity prompt to a more expensive model and simple tasks to more efficient models, which differs from most of the existing inference optimization methods for image generation. The empirical results demonstrated the efficacy of the proposed frameowk, which can achieve an average quality than a single model can achieve.

**Audience:**

Yes

**Audience Explanation:**

Compared to prior works in the image generation area, which mostly focus on parameter or step optimization, this method proposes a different solution that can employ various model architectures to achieve more diverse coverage and cost reduction. It successfully adapts the formulation and method from UMR to the text-to-image generation task and demonstrates its effectiveness with image specific quality metrics and estimators.

Therefore, I think it presents a solid innovation that can interest researchers in the related domains.

**Claims And Evidence:**

Yes

**Claims Explanation:**

The proposed framework is adapted from the formulation and proposition in Universal Model Routing for Efficient LLM Inference paper. It results in a plug-in estimator that can be applied to different model architectures and quality metrics.

The empirical evaluation in this paper is strong and demonstrates the efficacy of the proposed framework on two benchmark datasets (COCO, DiffusionDB) across multiple image quality metrics (CLIPScore, Sharpness, etc.). The results show that the proposed method can outperform static model baselines when using the same amount of TFLOPs, and on DiffusionDB it only needs 0.13% cost of a static model to achieve the same Sharpness.

**Requested Changes:**

I would recommend the authors bring the evaluation results with the FLUX.1-dev to the main paper as it demonstrates the method's efficacy on more recent architectures.

---

### Review · Reviewer_zXQn · 2025-12-23

**Summary Of Contributions:**

This paper proposed a method to improve quality/cost tradeoff of text-to-image generation by employing a router to select the optimal choice from a pool of generation functions, which could include different models and/or same model with different number of denoising steps. The optimization problem is basically framed as $argmax_m( quality(m) - \lambda* cost(m))$ where m is one of the options in the pool and lambda is a hyper-parameter depending on quality and cost metrics. Author further demonstrated two possible quality estimators, i.e. K-nearest neighbors-based and transformer-based, to plug into the optimization formula. Compared to using model inference to compute quality metrics, estimators only cost a fraction of FLOPs. Experimental results showed that the proposed method can achieve comparable or better quality under the same budget compared to a single fixed model.

**Audience:**

Yes

**Audience Explanation:**

text-image generation is a very important application and widely used nowadays.

**Broader Impact Concerns:**

No ethical implication concerns

**Claims And Evidence:**

Yes

**Claims Explanation:**

a decent amount of experimental data is provided. but some may need further clarifications to make it more convincing and clear. please see  Requested Changes below.

**Requested Changes:**

**1. Accuracy of quality estimator**

In Sec 5.3 second paragraph, the paper does not clearly distinguish the quality score predicted by the estimator from the one computed from real generated images using the option router selected. Take Fig. 3a, SDXL-only, transformer-based estimator (orange solid line) as an example. The only variable that router can control in this case is the number of denoising steps. For a given inference cost (in TFLOPs), the number of steps is determined accordingly, therefore, the router has no degree of freedom anymore. In other words, the orange solid line (the predicted score) should (ideally) match the actual quality scores calculated from SDXL generated images, i.e., red and brown stars. Compared to its counterpart in Fig 3b, apparently the SDXL estimator for CLIP score is not as accurate as the estimator for Sharpness score and tends to overestimate. While it's understandable that estimators do not always work perfectly, it is quite confusing to claim that "the proposed method" can outperform the baseline at a given inference cost in this case, because the proposed method can only choose baseline. Author may want to add a few sentences regarding estimator accuracy in this paragraph and consider clarifying whether a reported quality score is predicted by estimator or calculated from real (prompt, image) pairs for all the tables and figures.


**2. 100% Win rate in Table 1**

In Sec 5.4, "...Win Rate, defined as the fraction of trials that our router has higher test quality than the baseline..." Because router is choosing from all the possible options including baseline, intuitively there would be a certain probability that the router will choose the baseline model. For example, in Fig. 6 at ~260TFLOPs the router has ~45% of chance to pick SDXL, which means when compared to SDXL(-22), router will tie with baseline about half of the time. Therefore, 100% Win Rate in most of the cases in Table 1 doesn't seem very reasonable. Maybe author can elaborate a bit in Sec 5.4.


**3. Effect of hyper-parameter $\lambda$**

At the end of Sec 3.2, author intentionally stated that $\lambda$ is a hyper-parameter for users to tune. But as mentioned in Sec 3.3, this parameter ranges from 0 to inf, which makes it very difficult for readers to work with. It would be very helpful if the author can further discuss the effect and the behavior of this parameter, possibly provide a simple ablation study just like Sec 5.4 on Sample Size S.

---

### Author Response · Authors · 2026-02-22
**Authors to action editor and reviewers**

Dear action editor and reviewers,

We have submitted an updated manuscript and provided responses to all the reviewers' comments at the beginning of January 2026. May we ask whether there are further clarification or any information we can provide to help with the review process? Thank you for your service.

Thank you,
Authors

---

### Decision · Action_Editor_k98J · 2026-02-23

**Recommendation:** Accept as is

**Audience:**

Yes

**Audience Explanation:**

The findings of this paper will be of to interest researchers focused on text-to-image model optimization and deployment. The proposed learning-to-defer approach offers a reasonable solution to reducing inference costs without altering underlying architectures. Given the pervasive use of text-to-image diffusion models, this framework offers valuable insights into balancing generation quality and computational efficiency.

**Claims And Evidence:**

Yes

**Claims Explanation:**

Reviewers collectively agree that the claims are supported by strong empirical evidence. Authors conducted systematic and thorough evaluations on the COCO and DiffusionDB datasets across multiple image quality metrics to prove their method's efficacy. Furthermore, the reviewers noted that the inclusion of detailed deferral curves and ablation studies robustly validates the proposed cost-aware routing framework.